# Continual Learning on CLIP via Incremental Prompt Tuning with Intrinsic Textual Anchors

**Haodong Lu**[1,2]                                              *haodong.lu@unsw.edu.au*
**Xinyu Zhang**[3]                                              *xinyu.zhang@auckland.ac.nz*
**Kristen Moore**[2]                                            *kristen.moore@data61.csiro.au*
**Jason Xue**[2,4]                                              *jason.xue@data61.csiro.au*
**Lina Yao**[1]                                                 *lina.yao@unsw.edu.au*
**Anton van den Hengel**[4]                          *anton.vandenhengel@adelaide.edu.au*
**Dong Gong**[1]*                                               *dong.gong@unsw.edu.au*

[1] *School of Computer Science and Engineering, University of New South Wales*
[2] *Data61, CSIRO*
[3] *School of Computer Science, University of Auckland*
[4] *Australian Institute for Machine Learning (AIML), The University of Adelaide*

**Reviewed on OpenReview:** *https://openreview.net/forum?id=YJnjkzKq5Y*

## Abstract

Continual learning (CL) enables deep neural networks to acquire new knowledge over time while mitigating catastrophic forgetting of previously learned information. The powerful generalization ability of pre-trained models (PTMs), such as the Contrastive Language-Image Pre-training (CLIP) model, has inspired a range of CL methods targeting new and specialized tasks, further bridging the gap between PTMs and continual adaptation. Leveraging its multi-modal visual and textual representations, CLIP offers a natural paradigm for CL, where new tasks can be accommodated by incrementally learning lightweight parameters, particularly prompts. However, existing prompt-based CL methods for PTMs often rely on complex designs built upon specific assumptions, such as intricate regularization schemes for prompt pools, specialized routing mechanisms, or multi-stage incrementation processes. While these approaches improve performance, they frequently introduce additional—and possibly unnecessary—complexity, underutilizing CLIP's intrinsic capabilities. In this paper, we propose a concise CL approach for CLIP based on incremental prompt tuning that fully exploits its multi-modal structure and the stability of textual representations. Our method, Textual Prototype-guided Prompt Tuning (TPPT), introduces textual prototypes not merely as static classifiers, as in existing methods, but as stable anchors to guide the learning of visual prompts, thereby shaping the embedding space (*i.e.*, TPPT-V). We show that our bidirectional supervision strategy enables more effective learning of new knowledge while reducing forgetting. To further close the vision-language gap during CL, we activate the language branch and extend our approach to jointly optimize both visual and textual prompts (*i.e.*, TPPT-VT). We also introduce a relational diversity regularization on the textual anchors to prevent embedding space collapse and mitigate correlated forgetting. Extensive experiments and analyses demonstrate the effectiveness of our proposed approach, highlighting the benefits of leveraging CLIP's intrinsic guidance for continual adaptation. Code is available at https://github.com/jeff024/tppt.

## 1 Introduction

Continual learning (CL) focuses on learning from a sequence of data or tasks while accumulating knowledge over time and avoiding catastrophic forgetting of previously learned information. It aims to do so with

---

*D. Gong is the corresponding author.

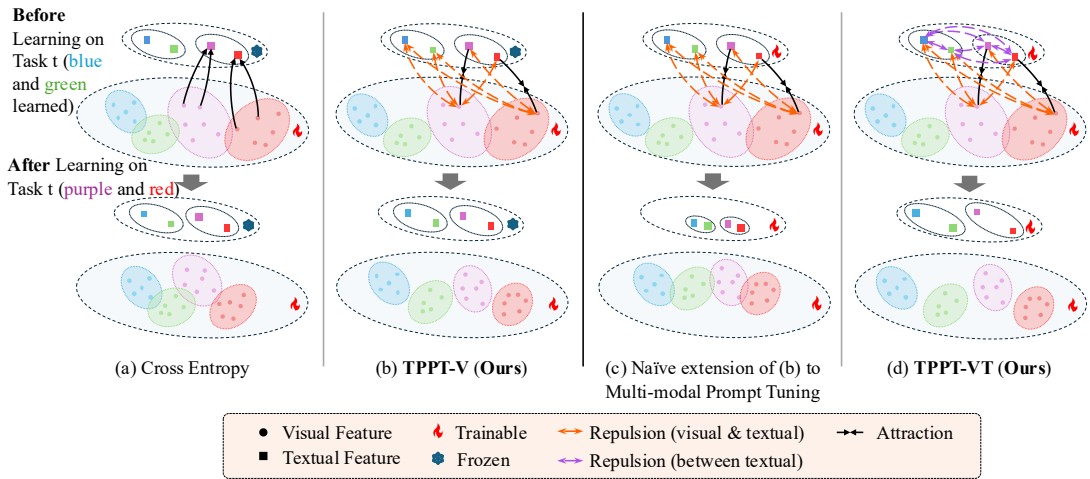

Figure 1: Conceptual illustrations of: (a) standard Cross-Entropy (CE), (b) our proposed TPPT-V, (c) a naïve multi-modal extension of TPPT-V, and (d) our proposed TPPT-VT. (a) Prior methods (Zhou et al., 2022c; Jia et al., 2022; Wang et al., 2022e;d; Smith et al., 2023; Wang et al., 2023; Zhou et al., 2023c) use CE loss to adapt PTMs, but suffer from representation drift (Gama et al., 2014; Lu et al., 2018), leading to forgetting. (b) TPPT-V introduces a textual prototypical contrastive loss to anchor visual features and mitigate drift. (c) A naïve extension that also tunes textual prompts may improve textual prototype quality but risks collapse to trivial solutions (Minderer et al., 2022; Kim et al., 2022; Khattak et al., 2023a; Liang et al., 2022). (d) TPPT-VT addresses this by regularizing multi-modal prompt learning with diversity constraints on textual prototypes.

minimal storage and computational overhead by eliminating the need to retain all previously encountered data (De Lange et al., 2021; Gomes et al., 2017; Mai et al., 2022). Our primary focus in this study is class-incremental learning (CIL), a representative CL setting where classification tasks arrive incrementally, each consisting of distinct, non-overlapping classes (Zhou et al., 2023a; Rebuffi et al., 2017). To mitigate catastrophic forgetting, various CL approaches have been explored, including experience replay (ER) methods (Luo et al., 2023; Aljundi et al., 2019b; Chaudhry et al., 2018a; Liu et al., 2020; Chaudhry et al., 2018b; Yan et al., 2022) and regularization-based techniques (Kirkpatrick et al., 2017; Aljundi et al., 2018; Zenke et al., 2017; Aljundi et al., 2019a; Jha et al., 2023), both of which have shown notable success.

Recent developments in pre-trained models (PTMs) based on the Transformer architecture (Vaswani et al., 2017) have demonstrated strong generalization capabilities and achieved state-of-the-art performance across various applications (Dosovitskiy et al., 2020; Devlin et al., 2018; Wortsman et al., 2022). Among these, multi-modal pre-trained models, such as Contrastive Language-Image Pre-training (CLIP) (Radford et al., 2021), jointly learn vision-language embedding spaces, effectively aligning visual and textual representations and enabling robust zero-shot generalization. To better adapt PTMs to downstream tasks, parameter-efficient fine-tuning methods such as prompt tuning (Zhou et al., 2022c; Yao et al., 2023; Khattak et al., 2023a;b) and adapters (Gao et al., 2023; Zhang et al., 2021; Sung et al., 2022) have been developed.

In CL settings, PTMs such as Vision Transformer (ViT) (Dosovitskiy et al., 2020) and CLIP (Radford et al., 2021) often serve as stable, "frozen" backbones for continually arriving downstream tasks (*e.g.*, in CIL). CL methods built on PTMs incrementally learn a fixed or dynamically expanding set of trainable parameters, such as soft prompts (Wang et al., 2022e;d; Lester et al., 2021; Liu et al., 2021; Smith et al., 2023) or adapters (Jha et al., 2024; Zhou et al., 2023b; McDonnell et al., 2024; Wang et al., 2024; Lu et al., 2025; Yu et al., 2024), to incorporate new tasks encountered in the data stream. CLIP, with its jointly trained vision and language representations, provides a natural foundation for CL (Jha et al., 2024; Wang et al., 2023), where textual embeddings can serve as pre-trained universal classifier heads for visual tasks. Due to their lightweight and non-intrusive nature, prompt-based methods have gained popularity in CL, particularly for CIL.

However, existing prompt-based continual learning methods with pre-trained models (including CLIP and ViT) often rely on task-specific prompt expansion (Smith et al., 2023; Wang et al., 2022d), complex regularization using external priors like orthogonality (Smith et al., 2023; Wang et al., 2022e; 2023), or intricate prompt routing schemes (Smith et al., 2023; Wang et al., 2022d;c). While these methods can boost performance, they often introduce additional and potentially unnecessary complexity, potentially underutilizing the intrinsic strengths of CLIP's multi-modal representations and the stability of its textual embeddings.

In this paper, we propose a concise CL approach for CLIP, called Textual Prototype-guided Prompt Tuning (TPPT), based on incremental prompt tuning without the need for complex regularization or architectural modifications. TPPT exploits CLIP's inherent multi-modal structure and the stability of its textual representations.

It accommodates new tasks by incrementally adding visual prompts (Jia et al., 2022) to the inputs of multiple multi-head self-attention (MSA) layers within the CLIP encoders. While prompt tuning has been widely used for CL with PTMs such as CLIP (Jha et al., 2024) and ViT (Smith et al., 2023; Wang et al., 2022d), prior works often introduce complexity through orthogonal regularization (Smith et al., 2023), probabilistic modeling (Jha et al., 2024), or handcrafted prompt selection (Smith et al., 2023; Wang et al., 2023).

In contrast, our method is simple and transparent, relying only on basic incremental and mixture mechanisms. This allows us to more clearly validate CLIP's inherent capabilities in CL.

We first present TPPT-V, which applies incremental prompt tuning only to the vision encoder of CLIP, guided by fixed textual embeddings of class labels. Prior works (Wang et al., 2023; Thengane et al., 2022; Ding et al., 2022; Zheng et al., 2023) typically treat these textual embeddings as static classifiers and use cross entropy (CE) loss to align vision features with corresponding textual embeddings. While this is effective when all classes are jointly trained, relying solely on CE leads to representation drift and interference (Gama et al., 2014; Lu et al., 2018) in CL due to partial class visibility and the model's unawareness of future tasks—resulting in forgetting (Nguyen et al., 2019; McCloskey & Cohen, 1989), as illustrated in Fig. 1(a) and Fig. 2(a).

As conceptualized in Fig. 1(a), this issue partly arises because CE loss focuses only on assigning each sample in isolation to a class label, ignoring how the embedding space evolves over time. We show that treating the fixed textual embeddings as prototype anchors (Li et al., 2020; Snell et al., 2017; Lu et al., 2024) and reintroducing contrastive learning supervision (Radford et al., 2021; Khosla et al., 2020; Grill et al., 2020) alleviates this problem. Although the textual prototypes remain fixed, the contrastive loss (specifically, the asymmetric CE formulation) serves as an additional regularizer, guiding visual embeddings away from unrelated prototypes and stabilizing the embedding space (Fig. 1(b)). In this way, CLIP's stable textual prototypes serve as anchors that help preserve prior knowledge by regularizing representation drift in the embedding space during CL.

To further enhance vision-language alignment in CL on downstream tasks, we go beyond vision-only tuning and enable learning of both visual and textual prompts. However, naïve joint prompt tuning can cause embedding space collapse (Minderer et al., 2022; Kim et al., 2022; Liang et al., 2022; Khattak et al., 2023a) (as shown in Fig. 1(c) and Fig. 2(b)), especially in CL, where the model can easily converge to trivial solutions (Minderer et al., 2022; Kim et al., 2022; Liang et al., 2022; Khattak et al., 2023a). To prevent this, we propose TPPT-VT (illustrated in Fig. 1(d)), which introduces relational diversity regularization on textual prompts, maintaining diversity among textual prototypes and mitigating collapse. By preserving the original contextual structure of textual representations, TPPT-VT improves performance over CL tasks.

**In summary, our contributions are:**

- We propose a concise CL approach (TPPT) for CLIP based on incremental prompt tuning, without relying on complex regularization, external priors, or additional architectural overhead. TPPT leverages CLIP's multi-modal nature and the stability of textual representations to explore its underutilized potential in CL.

- We propose two variants of TPPT. TPPT-V learns visual prompts guided by fixed textual prototypes, which serve as stable anchors in the embedding space—capturing both class semantics and past knowledge beyond mere classification. To further bridge the vision-text gap in CL, TPPT-VT jointly learns visual and textual prompts while applying a diversity regularization to prevent embedding collapse and forgetting.

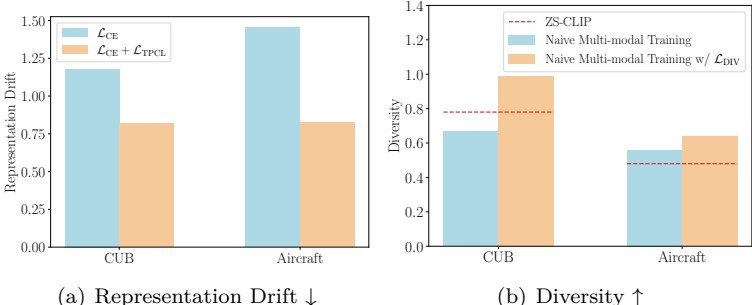

(a) Representation Drift ↓                    (b) Diversity ↑

Figure 2: Analysis of **(a)** representation drift (lower is better) and **(b)** feature embedding diversity (higher is better to prevent collapse). In **(a)**, we measure how far Task 1 embeddings deviate after learning all tasks. Training with CE loss alone leads to significant drift (Fig. 1(a)), whereas our proposed $\mathcal{L}_{\text{TPCL}}$ mitigates this (Fig. 1(b)). In **(b)**, we assess embedding diversity via average pairwise distances over incremental stages. The red dotted line represents the diversity of the pre-trained CLIP model, denoted as ZS-CLIP. Naïve multi-modal training reduces diversity, especially on CUB, resulting in lower scores and potential collapse (Fig. 1(c)). Our diversity-regularized approach alleviates this, as shown in Fig. 1(d).

- Comprehensive experiments and analyses that demonstrate TPPT's ability to mitigate catastrophic forgetting and improve adaptability to new tasks, achieving superior performance across multiple benchmarks.

## 2 Related Work

**Continual learning.** Experience replay (ER) based methods (Luo et al., 2023; Aljundi et al., 2019b; Chaudhry et al., 2018a; Liu et al., 2020; Chaudhry et al., 2018b; Yan et al., 2022; Tong et al., 2025) involve storing a subset of the original training data and replaying it during the training of new classes to remind the model of old knowledge. Knowledge distillation-based methods (Li & Hoiem, 2017; Rebuffi et al., 2017; Douillard et al., 2020) use a teacher-student approach, where the original teacher model helps in training the updated student model by transferring knowledge, thus preserving previously learned information. Parameter regularization methods (Kirkpatrick et al., 2017; Aljundi et al., 2018; Zenke et al., 2017; Aljundi et al., 2019a; Jha et al., 2023) add constraints to the learning process to retain important parameters from changing significantly, preserving prior knowledge. Dynamic networks (Yan et al., 2021; Wang et al., 2022a;b; Zhou et al., 2022a; Wang et al., 2024) adjust their architecture dynamically during learning, allowing different parts of the network to specialize in each task, thus balancing the learning of new classes with the preservation of old ones.

**Prompt tuning.** Prompt tuning, originally proposed in the Natural Language Processing (NLP) domain (Lester et al., 2021; Liu et al., 2022; 2021), involves adapting pre-trained foundational models for downstream tasks. This is achieved by introducing a small number of learnable embeddings at the input, known as prompt tokens. Prompt tuning technique have demonstrated their effectiveness at adapting to downstream vision tasks as well (Zhou et al., 2022c;b; Yao et al., 2023; Jia et al., 2022; Khattak et al., 2023a; Rasheed et al., 2023; Khattak et al., 2023b), by prepending learnable prompt tokens to either the text encoder (Zhou et al., 2022c;b; Yao et al., 2023) or the image encoder (Jia et al., 2022), or learning multi-modal prompts (Rasheed et al., 2023; Khattak et al., 2023a;b) that are integrated into both text and image encoders. More recently, multi-modal prompt tuning (Rasheed et al., 2023; Khattak et al., 2023a;b) has emerged, which involves learning deep multi-modal prompts that jointly adapt the vision and language branches of the CLIP model for enhanced performance.

**CL based on prompt tuning.** In light of the effectiveness of prompt tuning in adapting pre-trained models to downstream tasks, various prompt tuning-based methods (Wang et al., 2022e;d;c; 2023; Smith et al., 2023) have been developed for continual learning. Generally, these methods employ a set of prompts to preserve previously acquired knowledge, with each component designed to encode a segment of incoming knowledge. Specifically, L2P (Wang et al., 2022e) utilizes a fixed number of prompt sets and selects the top-K nearest prompts through query-key matching. DualPrompt (Wang et al., 2022d) innovates with both

global prompts and an incremental set of task-specific prompts, also employing query-key matching for selection. S-Prompts (Wang et al., 2022c) adopts a similar set of prompt designs, but they are specifically tailored for domain-incremental learning. CODA-P (Smith et al., 2023) learns an incremental set of visual prompts. It employs a weighted aggregation of visual prompts and learns the prompts relying on orthogonality regularization.

## 3 Proposed Method

### 3.1 Preliminaries

**Class-incremental continual learning.** Class-Incremental Learning (CIL) is a challenging scenario in the domain of CL (De Lange et al., 2021; Gomes et al., 2017; Mai et al., 2022), where a learning system is required to learn new classes over time without forgetting previously acquired knowledge (Lopez-Paz & Ranzato, 2017; Rebuffi et al., 2017). Formally, the learning process in CIL can be divided into a sequence of tasks $\mathcal{D} = \{\mathcal{D}^1, \mathcal{D}^2, \cdots, \mathcal{D}^T\}$, where each task $\mathcal{D}^t = \{(\mathbf{x}_i^t, y_i^t)\}_{i=1}^{n_t}$ contains tuples of the input sample $\mathbf{x}_i^t \in \mathcal{X}$ and its corresponding label $y_i^t \in \mathcal{Y}$. Following conventional CIL settings (Rebuffi et al., 2017; Hou et al., 2019; Wu et al., 2019), a small number of exemplars from the seen classes are selected as the exemplar set $\mathcal{E}$, and are used jointly with training data $\mathcal{D}_t$ of each task. CIL aims to develop a model that predicts the label $y$ given an unseen test sample $\mathbf{x}$ from arbitrary $T$ tasks.

**CLIP model.** In the CLIP model (Radford et al., 2021), the image and text encoders are represented as $f_\theta$ and $g_\psi$, respectively. For an input image $\mathbf{x} \in \mathbb{R}^{H \times W \times C}$, where $H$, $W$, $C$ represent the height, width, and channel of the image, it undergoes segmentation into patches and subsequent processing through several Transformer blocks. This process yields a latent visual feature representation $\mathbf{z} = f_\theta(\mathbf{x}) \in \mathbb{R}^D$. $D$ represents the dimension of the embedding space. The corresponding class label $y$ is integrated into hand-crafted prompts like "a photo of a [CLS]" to obtain text input $\mathbf{t}$, where [CLS] denotes the class name associated with class label $y$. This prompt is then encoded by the text encoder $g_\psi$ to produce the text embedding $\mathbf{w} = g_\psi(\mathbf{t}) \in \mathbb{R}^D$. The probability of a given image $\mathbf{x}$ being classified into class $y_i \in \{1, 2, \cdots, C\}$ is calculated as follows:

$$p(y_i|\mathbf{x}) = \frac{\exp(\mathtt{sim}(\mathbf{z} \cdot \mathbf{w}_{y_i}/\tau)}{\sum_{c=1}^C \exp(\mathtt{sim}(\mathbf{z} \cdot \mathbf{w}_{y_c}/\tau))}, \tag{1}$$

where $\mathtt{sim}(\cdot)$ denotes the cosine similarity function, and $\tau$ is the temperature parameter.

**Prompt tuning for CLIP.** Prompt tuning has emerged as a highly effective and efficient technique for adapting pre-trained foundational models to various downstream tasks. A range of methods (Zhou et al., 2022c; Yao et al., 2023; Rasheed et al., 2023; Khattak et al., 2023a;b) have been developed for adapting VLM. Specifically, a visual prompt $\mathbf{P}_{v,1} \in \mathbb{R}^{L_v \times D}$ and a textual prompt $\mathbf{P}_{t,1} \in \mathbb{R}^{L_t \times D}$ are prepended to the the visual and textual input tokens, steering the intermediate hidden states of the first layer of the model, and subsequently influence the final prediction. Here, $L_v$ and $L_t$ denote the lengths of the visual and textual prompts. The prompted visual and textual features are then denoted as $\tilde{\mathbf{z}} = \tilde{f}_\theta(\mathbf{x})$ and $\tilde{\mathbf{w}} = \tilde{g}_\psi(\mathbf{t})$, respectively. Here, $\tilde{f}_\theta$ and $\tilde{g}_\psi$ represent the encoders augmented with prompt tokens for image and text, respectively.

### 3.2 Overview

We present the framework of our methods in Fig. 3. Previous CLIP-based CL methods (Wang et al., 2022c; 2023; Zhou et al., 2023c) usually treat the text encoder as a pre-trained classifier, overlooking the rich representation capability of the language modality. In Sec. 3.3 we introduce **TPPT-V**, which guides the learning of incremental visual prompts with static textual prototypes. Since learning with static textual prototypes may be limited by the quality of prototypes, in Sec. 3.4 we further propose **TPPT-VT** to learn textual prompts for higher quality and representative textual prototypes. Our approach encourages textual diversity to better regulate the learning process.

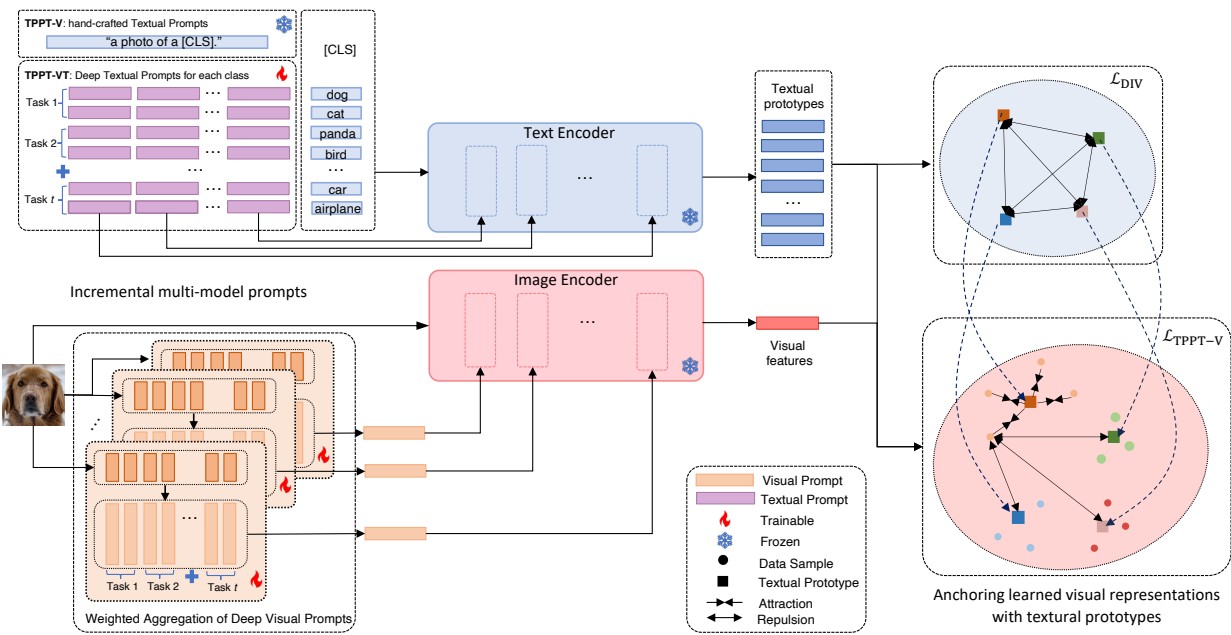

Figure 3: The overall framework of our 2 proposed methods. (1) The learned visual representations are guided by static textual prototypes (TPPT-V). We alleviate the forgetting issue by guiding visual representations with consistent textual prototypes, preventing drift of representations in the embedding space. (2) To improve upon the static textual prototypes, we propose to learn textual prompts for prototypes (TPPT-VT), and regulate the learning process by encouraging diversity. (3) Benefiting from the textual prototype anchors, our proposed methods remain simple yet effective, unlike previous methods that use delicate, complex designs.

## 3.3 Incremental Visual Prompt Tuning with Textual Prototypes

**Incremental visual prompt tuning.** Prompt-based methods in CL have been proven to be highly effective in mitigating catastrophic forgetting, primarily by utilizing a fixed or incremental set of prompts (Wang et al., 2022e;d; Smith et al., 2023). Building on this, we employ a straightforward method to prepend an incremental set of visual prompts on multiple Transformer layers (Jia et al., 2022; Rasheed et al., 2023; Khattak et al., 2023b). These prompts are denoted as $\mathbf{P}_{v,l} = \{\mathbf{P}_{v,l}^1, \mathbf{P}_{v,l}^2, \cdots, \mathbf{P}_{v,l}^M\}$ for each Transformer layer $l$ ranging from the first layer up to the $d_v$-th layer of the vision encoder, where $M$ is the total number of prompts, and $d_v$ is a hyperparameter indicating the depth of Transformer layers at which prompts are learned.

During both the training and testing phases, instance-specific visual prompts are generated for each input image, using a weighted aggregation mechanism. The prompts used for a given input image $\mathbf{x}$ at the $l$-th Transformer layer during training and inference are computed as follows:

$$\mathbf{P}_{v,l} = \sum_{m=1}^{M} \alpha_l^m \mathbf{P}_{v,l}^m, \tag{2}$$

where the aggregation weights $\alpha_l = \mathrm{LN}_l\big(q(\mathbf{x})\big) \in \mathbb{R}^M$ are produced by a lightweight trainable linear layer applied at each layer with trainable prompts to obtain the affinities for the $M$ candidate prompts, in contrast to the complicated attended query-key matching with additional learnable attention vectors used in (Smith et al., 2023). Here, the query function $q(\cdot) = f_\theta(\cdot) \in \mathbb{R}^D$ refers to the pre-trained image encoder (Wang et al., 2022e;d; Smith et al., 2023). Our concise formulation adopts the widely used query–key prompt selection/aggregation mechanism (Wang et al., 2022e;d; Smith et al., 2023) and simplifies prior designs by omitting incremental learnable attention vectors and orthogonal regularization (Smith et al., 2023). The prompt tuning process is guided by consistent and stable knowledge derived from previously learned information through textual prototypes during incremental stages.

**Anchoring the past with textual prototypes.** Existing CLIP-based CIL methods (Wang et al., 2023; Thengane et al., 2022) often utilize the text embeddings merely as a pre-trained classifier, thereby not fully exploiting the rich representational capacity of the language modality. To this end, we propose **TPPT-V** to guide the learning of incremental visual prompts through the use of static textual prototypes as anchor points in the embedding space. TPPT-V mitigates catastrophic forgetting by consistently ensuring that visual representations remain closely aligned and compact around corresponding static textual prototypes.

Specifically, our textual prototypes are derived from text embeddings, obtained through hand-crafted prompt templates with class names from OpenAI[1] whenever available. For datasets where prompt templates are not provided, we apply the basic template of "a photo of a [cls]". To enhance the alignment (Wang & Isola, 2020) between visual features and their corresponding textual prototypes, we propose the Textual Prototypical Contrastive Loss (Li et al., 2020; Cui et al., 2023; Li et al., 2023). This approach leverages the robust pre-trained representational capacity of these textual prototypes while preventing visual features from being assigned to the classes previously learned. It also effectively prevents the drift of previously learned semantics throughout the incremental learning phase (Fig. 5). We denote the Textual Prototypical Contrastive Loss as:

$$\mathcal{L}_{\text{TPCL}} = \frac{1}{C} \sum_{c=1}^{C} \sum_{i=1}^{N} -\log \frac{\mathbb{1}(y_i = c) \exp(\texttt{sim}(\mathbf{w}_c \cdot \tilde{\mathbf{z}_i}/\tau))}{\sum_{i'=1}^{N} \exp(\texttt{sim}(\mathbf{w}_c \cdot \tilde{\mathbf{z}_{i'}}/\tau))}, \tag{3}$$

where $\mathbb{1}(\cdot)$ is the indicator function that selects samples corresponding to the given textual prototype. $\tilde{\mathbf{z}} = \tilde{f}_\theta(\mathbf{x})$ is the visual embedding obtained through the prompted image encoder, and $\mathbf{w}_c$ is the text embedding for class $c$, acting as a static textual prototype. $N$ denotes the number of training examples in the current mini-batch, and $C$ is the number of classes already seen. At incremental step $t$, we construct the textual prototype set using only the class labels that have been introduced up to the current task. Text embeddings for future classes are not instantiated or used in any component.

**Training objectives.** The overall training objective for **TPPT-V** is formally denoted as:

$$\mathcal{L}_{\text{TPPT-V}} = \mathcal{L}_{\text{CE}} + \mathcal{L}_{\text{TPCL}}, \tag{4}$$

where $\mathcal{L}_{\text{CE}}$ represents the cross-entropy loss between predicted probabilities and ground truth. All loss terms are computed with the prompted image encoder $\tilde{f}_\theta$ and the frozen pre-trained text encoder $g_\psi$. This loss shares a similar form with the pre-training loss of the CLIP model (Radford et al., 2021), but differs in purpose. While CLIP focuses on the one-to-one mapping between an image and its own caption to form a broad representation space, our method aims to cluster image embeddings to be compact around their corresponding class text embeddings, from the perspective of CL to mitigate forgetting.

### 3.4 Incremental multi-modal prompt tuning with regularized textual prototypes

**Incremental textual prompt tuning.** Although static textural prototypes from hand-crafted text prompts of classes effectively serve as stable anchors in TPPT-V, the effectiveness can be limited by the gap between pre-trained CLIP and specific downstream tasks (*e.g.*, fine-grained classification), as well as the inherent discrepancy between the textual and visual embeddings in CLIP (Liang et al., 2022). To address these limitations, we introduce **TPPT-VT** to further incorporate multi-model prompt tuning using learnable soft text prompts. Similar to the design of visual prompts in TPPT-V, we use an incremental set of textual prompts for each Transformer layer $l$, ranging from the first layer up to the $d_t$-th layer of the text encoder, where $d_t$ is set as a hyperparameter indicating the depth of Transformer layers at which prompts are learned. We denote these prompts as $\mathbf{P}_{t,l} = \{\mathbf{P}_{t,l}^1, \mathbf{P}_{t,l}^2, \cdots, \mathbf{P}_{t,l}^C\}$, where each class is assigned to a unique textual prompt, and $C$ represents the total number of classes encountered thus far. By using unique prompts for each class, TPPT-VT further enhances the fixed textual prototypes in TPPT-V, making them more representative.

**Regularizing textual prompt tuning.** While guiding the model with textual prototypes derived from hand-crafted prompts has demonstrated superior performance, the joint training of textual and visual prompts introduces new challenges. One critical issue is the model collapse, *i.e*, both branches of the joint-embedding model produce nearly identical and constant output vectors (Jing et al., 2021; Bardes et al., 2021) (as

---

[1]https://github.com/openai/CLIP

illustrated in Fig. 6). This collapse compromises the model's ability to distinguish between diverse inputs and effectively learn meaningful representations. In addition, naïve joint training can adversely affect the uniformity of the pre-trained model (Wang & Isola, 2020). Uniformity is a desirable property for joint-embedding models, ensuring that feature vectors are uniformly distributed roughly on the unit hypersphere. This uniform distribution is essential for preserving the maximum amount of information from the data.

To address these issues, we introduce the textual prototype diversity loss, which regularizes the model by explicitly enforcing textual prototypes to be uniformly distributed on the unit hypersphere. This is achieved by promoting high pairwise distances among textual prototypes (Wang & Isola, 2020; Cho et al., 2023), formally defined as follows:

$$\mathcal{L}_{\text{DIV}} = \log \sum\nolimits_{m,n \in [C], m \neq n} \exp(-\|\tilde{w}_m - \tilde{w}_n\|_2^2), \tag{5}$$

where $[C]$ represents the index set of the already-seen $C$ classes, and $m \neq n$ indicates that the computation of pairwise distances excludes the self comparisons.

**Training objectives.** The overall training objective for **TPPT-VT** is formally given by:

$$\mathcal{L}_{\text{TPPT-VT}} = \mathcal{L}_{\text{TPPT-V}} + \alpha \mathcal{L}_{\text{DIV}}, \tag{6}$$

where $\alpha$ indicates the weight of textual diversity loss, and all loss terms are computed with the prompted image encoder $\tilde{f}_\theta$ and the prompted text encoder $\tilde{g}_\psi$.

## 4 Experiments

### 4.1 Experiment Settings

**Datasets.** In line with common CIL benchmarks (Rebuffi et al., 2017; Wang et al., 2022e;d; Smith et al., 2023), we evaluate the performance using widely used datasets CIFAR100 (Krizhevsky et al., 2009), ImageNet-R (Deng et al., 2009), TinyImageNet (Le & Yang, 2015), and CUB200 (Wah et al., 2011). To further validate the efficacy of our approach, we extend our evaluation to include two additional datasets, FGVCAircraft (Maji et al., 2013) and Stanford Cars (Krause et al., 2013), following the popular VLM methods (Zhou et al., 2022c; Yao et al., 2023; Khattak et al., 2023a;b). Specifically, we split the training datasets into 10 tasks, resulting in 10 classes per task for CIFAR100, Aircraft, and Stanford Cars, 20 classes per task for ImageNet-R, TinyImageNet, and CUB200. We sample 20 exemplars per class as our replay buffer as in (Rebuffi et al., 2017).

**Training details.** For a fair comparison, we use the same backbone model CLIP with ViT-B/16, using pre-trained weights from (Ilharco et al., 2021; Radford et al., 2021). We train all methods with a batch size of 64 for 5 epochs, using SGD with momentum for optimization. The learning rate starts from 0.1 and decays with cosine annealing. The visual prompt length $L_v$ and the textual prompt length $L_t$ are set to 4, and the depth for both visual prompts $d_v$ and textual prompts $d_t$ are set to 12. The number of prompts per task is set to 10. The loss weight $\alpha$ is set to 1 by default.

**Evaluation metrics.** To demonstrate the effectiveness of our approach, we report two commonly used metrics (Rebuffi et al., 2017): (1) the final accuracy $\mathcal{A}_T$ after learning all the $T$ incremental stages, (2) the average accuracy $\bar{\mathcal{A}} = \frac{1}{T} \sum_{t=1}^{T} \mathcal{A}_t$ of the individual accuracies across the incremental stages.

### 4.2 Experimental Results

**Comparisons with prompt-based methods.** In Table 1, we evaluate the final and average accuracies across various datasets, comparing current state-of-the-art methods. Additionally, we illustrate the performance across incremental stages in Fig. 4, and provide evaluations of multiple independent runs with varied orders of the incoming classes in Appendix C. Our proposed methods, **TPPT-V** and **TPPT-VT**, exhibit a remarkable improvement over previous approaches. In contrast to previous prompt tuning methods like CoOp and VPT, we learn an incremental set of prompts, effectively mitigating the issue of incoming tasks overwriting

Table 1: Average and final accuracy of different methods. All methods are trained with identical exemplar sizes, utilizing the CLIP model with a ViT-B/16 backbone using the same pre-trained weight from LAION-400M (Ilharco et al., 2021). $\bar{\mathcal{A}}$ denotes the average test accuracy over the incremental stages, $\mathcal{A}_T$ denotes the test accuracy at the last incremental stage. The **best** and **second best** performances are indicated in **red** and **blue**, respectively.

| Methods | CIFAR100 | | ImageNet-R | | CUB | | Aircraft | | Cars | |
|---|---|---|---|---|---|---|---|---|---|---|
| | $\bar{\mathcal{A}}$ | $\mathcal{A}_T$ | $\bar{\mathcal{A}}$ | $\mathcal{A}_T$ | $\bar{\mathcal{A}}$ | $\mathcal{A}_T$ | $\bar{\mathcal{A}}$ | $\mathcal{A}_T$ | $\bar{\mathcal{A}}$ | $\mathcal{A}_T$ |
| ZS-CLIP | 81.38 | 71.26 | 82.93 | 76.67 | 76.01 | 64.50 | 26.66 | 17.22 | 82.60 | 76.37 |
| CoOp | 83.37 | 73.36 | 82.40 | 76.20 | 77.34 | 68.70 | 44.26 | 39.87 | 89.73 | 84.91 |
| VPT-CL | 87.11 | 78.21 | 84.60 | 78.97 | 80.81 | 69.89 | 35.52 | 25.50 | 93.71 | 91.49 |
| L2P | 76.42 | 66.21 | 75.73 | 67.22 | 79.23 | 68.54 | 55.06 | 44.88 | 83.81 | 72.44 |
| DualPrompt | 79.07 | 70.06 | 78.47 | 70.82 | 83.21 | 74.94 | 50.93 | 46.53 | 85.30 | 74.35 |
| CODA-P | 83.53 | 72.20 | 74.45 | 65.88 | 83.36 | 76.12 | 54.51 | 45.06 | **96.16** | 92.17 |
| PROOF | 86.70 | **79.05** | **85.34** | 80.10 | 84.93 | 79.43 | 61.00 | 53.59 | 93.26 | 89.84 |
| TPPT-V (Ours) | **88.28** | **80.81** | **85.62** | **80.75** | **85.77** | **80.28** | **63.47** | **54.85** | 95.26 | **94.57** |
| TPPT-VT (Ours) | **87.20** | 78.98 | 84.87 | **81.02** | **86.60** | **81.55** | **69.34** | **61.99** | **96.83** | **95.37** |

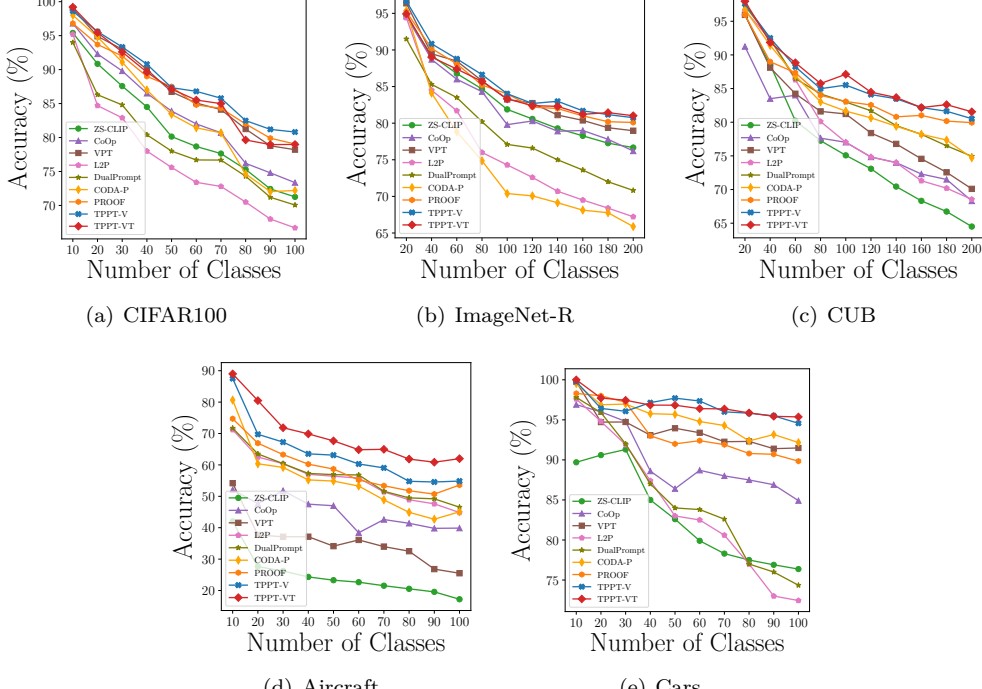

Figure 4: Experiment results across incremental stages. All methods are trained with the same exemplar size using the same pre-trained weights and backbone model of CLIP with ViT-B/16.

previously learned parameters. Compared with prior CIL methods, it steers the learning of visual prompts using fixed textual prototypes, providing reliable anchor points in the embedding space. TPPT-VT can perform the best (and better than TPPT-V) consistently on the datasets with fine-grained classification characteristics, *i.e.*, CUB, Aircraft, and Cars, which indicates that the textual prompt tuning helps to fill the gap on downstream tasks that have more specific characteristics. For the normal datasets, *e.g.* CIFAR100 and ImageNet-R, text prompt tuning may introduce training difficulties, influencing the performance of TPPT-VT. In Table 2, we extend the comparisons to include two related methods DIKI and CPP, with further details in Appendix A.2.

**Comparisons with fine-tuning and adapter/LoRA-Based methods.** In Table 2, we compare our method against fine-tuning-based approaches such as ZSCL (Zheng et al., 2023) and more recent

Table 2: Comparison with prior methods on CIFAR-100 and Tiny-ImageNet using a 10-task split following the experiment settings of (Zheng et al., 2023; Yu et al., 2024). All methods utilize CLIP (ViT-B/16) with pre-trained weights from OpenAI (Radford et al., 2021).

| Methods | CIFAR100 | | TinyImageNet | |
|---|---|---|---|---|
| | $\bar{\mathcal{A}}$ | $\mathcal{A}_T$ | $\bar{\mathcal{A}}$ | $\mathcal{A}_T$ |
| ZS-CLIP | 74.47 | 65.92 | 69.55 | 65.59 |
| ZSCL | 82.15 | 73.65 | 78.61 | 71.62 |
| InfLoRA | **85.66** | 75.09 | 79.88 | 74.42 |
| DIKI | 84.43 | 76.14 | 81.29 | 75.66 |
| CPP | 84.65 | 75.57 | 77.92 | 72.22 |
| CLAP4CLIP | 82.09 | 76.77 | 80.90 | 74.44 |
| MoE-Adapter | 85.15 | 75.98 | 80.23 | 76.35 |
| TPPT-V (Ours) | 83.88 | **77.85** | **83.18** | **77.51** |
| TPPT-VT (Ours) | **85.72** | **77.75** | **82.41** | **78.08** |

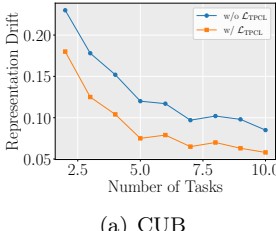

(a) CUB

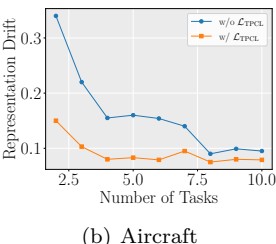

(b) Aircraft

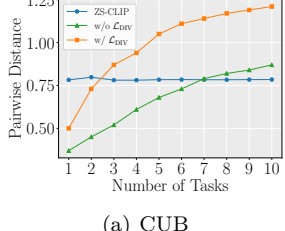

(a) CUB

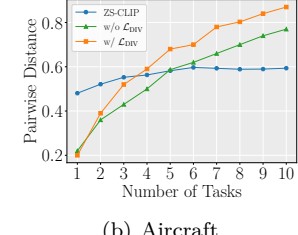

(b) Aircraft

Figure 5: Representation drift ↓ (lower is better) across incremental stages for (a) CUB and (b) Aircraft. Representation drift measures the divergence in the means of visual features for each class compared to the last incremental stage.

Figure 6: Pairwise distance ↑ (higher is better) between visual class means across incremental stages for (a) CUB and (b) Aircraft. A higher pairwise distance indicates the model has better learning capacity and is less prone to collapse.

adapter/LoRA (Hu et al., 2021)-based methods, including InfLoRA (Liang & Li, 2024), MoE-Adapter (Yu et al., 2024), and CLAP4CLIP (Jha et al., 2024). Despite utilizing additional reference data (Zheng et al., 2023), fine-tuning-based methods suffer from rapid forgetting due to full model updates and are computationally expensive. Adapter-based methods, on the other hand, struggle to maintain vision-text alignment across incremental learning stages, leading to suboptimal knowledge retention. CLAP4CLIP adopts probabilistic fine-tuning with adapters and mixtures of task logits, which leaves the image–text embedding space under-constrained. In contrast, our approach leverages textual prototypes as stable anchors to guide the learning of incremental visual and multi-modal prompts. Our proposed method effectively mitigates representation drift, enhances cross-task consistency, and significantly improves continual learning performance while maintaining computational efficiency. Additional comparisons with the fine-tuning–based SPU method (Zhang et al., 2024) are provided in Appendix Table 10.

### 4.3 Discussion

**Learning with textual prototypes prevents representation drift.** Catastrophic forgetting in continual learning is often attributed to representation drift (Gama et al., 2014; Lu et al., 2018), a phenomenon where the statistical properties of the target variable, which the model aims to predict, undergo changes during incremental learning. In Fig. 5, we measure representation drift during the incremental learning process by computing the divergence in the means of visual features for each class, relative to the class means from the previous incremental stage. We find that the phenomenon of representation drift (Gama et al., 2014; Lu et al., 2018) is severe, especially for the first few incremental stages. By enforcing visual features to be

Table 3: Effectiveness of Textual Prototypical Contrastive Loss in prompt tuning methods like VPT-CL and TPPT-V (Ours), on both CIL ($\bar{\mathcal{A}}$ and $\mathcal{A}_T$) and Joint Training ($\mathcal{A}_{Joint}$).

| Methods | | CUB | | | Aircraft | | |
|---|---|---|---|---|---|---|---|
| | | $\bar{\mathcal{A}}$ | $\mathcal{A}_T$ | $\mathcal{A}_{Joint}$ | $\bar{\mathcal{A}}$ | $\mathcal{A}_T$ | $\mathcal{A}_{Joint}$ |
| VPT-CL | w/o $\mathcal{L}_{\text{TPCL}}$ | 80.81 | 69.89 | 71.97 | 35.52 | 25.50 | 30.36 |
| | w/ $\mathcal{L}_{\text{TPCL}}$ | **81.42** | **71.57** | **73.37** | **41.73** | **30.57** | **30.45** |
| TPPT-V | w/o $\mathcal{L}_{\text{TPCL}}$ | 83.78 | 76.97 | 79.64 | 60.53 | 53.20 | 57.19 |
| | w/ $\mathcal{L}_{\text{TPCL}}$ | **85.77** | **80.28** | **79.77** | **64.58** | **54.85** | **59.35** |

Table 4: Ablation on Textual Diversity.

| Methods | | CUB | | Aircraft | |
|---|---|---|---|---|---|
| | | $\bar{\mathcal{A}}$ | $\mathcal{A}_T$ | $\bar{\mathcal{A}}$ | $\mathcal{A}_T$ |
| TPPT-VT | w/o $\mathcal{L}_{\text{DIV}}$ | 85.18 | 78.24 | 67.99 | 59.44 |
| | w/ $\mathcal{L}_{\text{DIV}}$ | **86.60** | **81.55** | **69.34** | **61.99** |

compact around their fixed textual prototypes through our proposed textual prototypical contrastive loss, we effectively mitigate catastrophic forgetting by significantly reducing representation drift.

**Regularized textual prototypes encourage uniformity and prevent collapse.** We quantify the pairwise distances between class means of visual features across different incremental stages in Fig. 6. ZS-CLIP with pre-trained weight exhibits consistent uniformity (Wang & Isola, 2020; Cho et al., 2023) and is less prone to collapse (Jing et al., 2021; Bardes et al., 2021), due to the massive amount of data seen during pre-training. However, challenges arise when textual and visual prompts are learned jointly. In the initial incremental stages, the model tends to exhibit a tendency towards collapse, characterized by class means clustering too closely. The proposed diversity regularization effectively promotes uniformity of the model and prevents collapse.

### 4.4 Ablation Studies

**Effectiveness of textual prototypical contrastive loss.** In Table 3, we demonstrate the effectiveness of our proposed textual prototypical contrastive loss on prompt tuning methods, including VPT-CL and our proposed method TPPT-V, across CUB and Aircraft datasets under both Class-Incremental Learning and Joint Training settings. For Joint Training, where models are exposed to all classes simultaneously during training, a good decision boundary between classes can be achieved using CE alone, while learning with textual prototypical contrastive loss slightly improves the performance.

Most importantly, for CIL, during the early incremental stages, models initially encounter only a tiny fraction of the total knowledge to be learned. The model has no idea what it will encounter in the future; thus, learning current data does not take future data into consideration. However, when learning new data, some embedding space is indeed needed to accommodate upcoming tasks (as illustrated in Fig. 1(a)), which causes representation drift (as shown in Fig. 5) of previously learned data. The use of textual prototypical contrastive loss helps mitigate this issue by maintaining more stable and meaningful embeddings throughout the incremental learning process.

**Effectiveness of Textual Diversity Regularization.** Table 4 ablates our diversity loss, which enforces uniform textual prototypes on the unit hypersphere to boost per-class retention. Fig. 7 demonstrates robustness across regularization weights, performance peaks at 0.8 and we adopt 1 by default for simplicity.

**Number of prompts.** In Fig. 8, we evaluate the impact of varying the number of prompts per task on the performance of our methods. We conducted tests with our methods using a range of prompts per task, specifically {1, 2, 3, 4, 5, 10, 20, 50}. Our findings reveal that for TPPT-V on the CUB dataset, using only a single prompt per task is inadequate. Both TPPT-V and TPPT-VT exhibit favorable performance when utilizing at least 5 prompts per task. Consequently, we opt for 10 prompts per task by default, ensuring an effective trade-off between the training budget and performance.

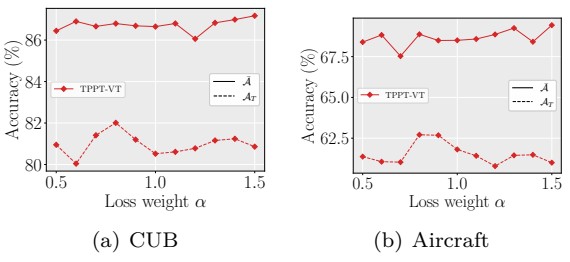
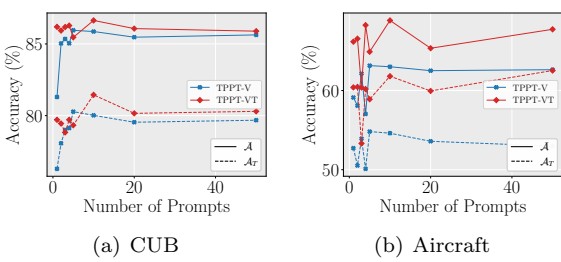

(a) CUB (b) Aircraft

Figure 7: Effect of the textual diversity loss weight on (a) CUB and (b) Aircraft.

(a) CUB (b) Aircraft

Figure 8: Ablation studies on the number of prompts per task for (a) CUB and (b) Aircraft.

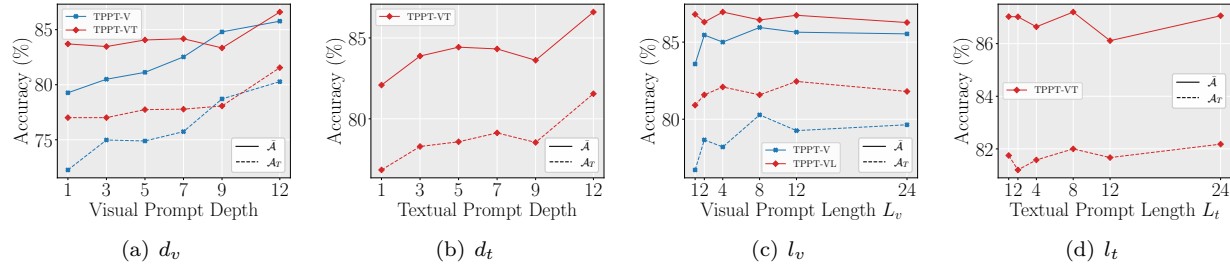

(a) $d_v$ (b) $d_t$ (c) $l_v$ (d) $l_t$

Figure 9: Ablations on multi-modal prompt formulation. Sub-figures (a) and (b) analyze the depth of visual and textual prompts, while (c) and (d) investigate the length of visual and textual prompts.

**Prompt formulation.** In Fig. 9, we ablate the design of visual and textual prompts utilized in our proposed methods. Overall, both TPPT-V and TPPT-VT show improved performance with deeper prompts, where prompts are embedded into more advanced layers of Transformer blocks. Regarding the length of prompts, extended visual prompts benefit the prediction accuracy for both methods, while the performance of these methods appears relatively insensitive to the length of textual prompts.

**Discussion on computation cost.** Similar to our design, CODA-P, which also uses incremental visual prompts, introduces 5.68M parameters and reaches an average final accuracy of 70.29. Without the need for additional learnable attention vectors, TPPT-V achieves 78.25% with only 4.3 M parameters, and TPPT-VT further elevates this to 80.07% with an extra 2.46 M parameters. More Details in Appendix Table 13.

## 5 Conclusion

Our work introduces TPPT-V and TPPT-VT, which leverage textual prototypes to guide CLIP visual prompts, further aligning text and vision via learned textual prompts and diversity regularization, respectively. Extensive experiments show that our simple, efficient design substantially reduces forgetting while improving adaptation to new tasks, representing a notable advance in CL.

**Limitations and Future Work.** We introduce a general supervision strategy for vision–language models, demonstrated mainly using CLIP. Our proposed method is generally applicable to any models and settings, and we leave evaluation on other architectures and experimental settings to future work.

### Broader Impact Statement

This work represents fundamental research in machine learning, aiming to enable pre-trained foundation models to extend their knowledge base to downstream tasks without compromising their original strong pre-trained knowledge. The proposed research has the potential to be applied to various real-world applications, including computer vision systems, robotics, embodied AI, and autonomous driving. A potential negative social impact involves privacy concerns, as our methods utilize a replay buffer containing previous data samples, which may not be ideal for privacy-sensitive applications. However, this negative effect is minimal, as our methods still perform well even under restricted memory sizes, thereby reducing the reliance on extensive data storage.

**Acknowledgments**

This work was partially supported by the ARC DECRA Fellowship (DE230101591) awarded to D. Gong. H. Lu is affiliated with CSIRO Data61 through a PhD scholarship from CSIRO and acknowledges the support of the Google PhD Fellowship.

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

# A More Details of the Proposed Method

## A.1 Algorithm Details

We illustrate the algorithm details of TPPT-VT in Algorithm 1. The training pipeline of TPPT-V follows the same structure as TPPT-VT, except that the textual features are obtained with the frozen text encoder without textual prompts, and the training objectives for TPPT-V are detailed in Equation 4.

---

**Algorithm 1:** Training process of the proposed TPPT-VT

---

**Input:** Pre-trained image encoder $f_\theta$, pre-trained text encoder $g_\psi$, number of tasks $T$, training epochs $E$,
  training set $\{\{(\mathbf{x}_i^t, y_i^t)\}_{i=1}^{n_t}\}_{t=1}^T$, visual prompts $\boldsymbol{P}_v$, linear layer for obtaining weights for visual prompts
  $\mathrm{LN}(\cdot)$, textual prompts $\boldsymbol{P}_t$

**Initialize:** $\boldsymbol{P}_v, \mathrm{LN}(\cdot), \boldsymbol{P}_t$

**for** $t = 1, \cdots, T$ **do**

   **for** $e = 1, \cdots, E$ **do**

      Draw a mini-batch $B = \{(\mathbf{x}_i^t, y_i^t)\}_{i=1}^l$

      **for** $(\mathbf{x}, y) \in B$ **do**

         Generate query feature $q(\mathbf{x})$ and visual prompts aggregation weights via $\alpha = \mathrm{LN}(q(\mathbf{x}))$

         Aggregate the visual prompts $\boldsymbol{p}_v$ for given $\mathbf{x}$ via Equation 2

         Select the textual prompts $\boldsymbol{p}_t^y$ corresponding to class $y$

         Obtain prompted visual feature $\tilde{\mathbf{z}}$ and textual feature $\tilde{\mathbf{w}}$

         Calculate per sample loss via Equation 6

      Update $\boldsymbol{P}_v, \mathrm{LN}(\cdot), \boldsymbol{P}_t$ by backpropagation

---

## A.2 More Discussions with Related Works

DIKI (Tang et al., 2024) and CPP (Li et al., 2024) tune prompts under primarily intra-visual supervision (calibration in DIKI; prototype steering and routing in CPP), which can leave cross-modal alignment under-constrained. TPPT anchors learning to CLIP text prototypes (fixed/learnable) and optimizes a cross-modal TPCL with text-side diversity, stabilizing the image–text space. CLAP4CLIP (Jha et al., 2024) (adapters with probabilistic logit mixtures) and LoRA/MoE-Adapter variants (Liang & Li, 2024; Yu et al., 2024) struggle to maintain vision-text alignment across incremental learning stages, leading to suboptimal knowledge retention. SPU (Zhang et al., 2024) selects a subset of important MLP layers via gradient-based scoring and updates them directly, which introduces greater computational overhead than optimizing only a small set of prompt tokens. In contrast, TPPT keeps the backbone frozen and anchors learning to CLIP text prototypes (fixed and learnable), optimizing a cross-modal TPCL with text-side diversity to stabilize the image–text embedding space, and TPPT outperforms these methods (Table 2 and Appendix Table 10).

TreeProbe (Zhu et al., 2023) pairs a fixed CLIP zero-shot head with an exemplar probe and uses tree-style gating to choose between stored exemplars and pre-trained knowledge for open-vocabulary increments. AnytimeCL (Zhu et al., 2024) emphasizes update scheduling and memory compression to support single-sample, streaming updates. CGIL (Frascaroli et al., 2024) performs generative latent replay by training a VAE in CLIP's feature space to synthesize past examples. These directions prioritize buffer usage/quality, scheduling/compression, or replay—orthogonal to our method, which targets representation learning via textual anchors and cross-modal alignment.

# B Additional Experimental Details

We provide additional details of our experiments in this appendix section.

### B.1 Datasets

For a thorough evaluation of our methodologies, we compare the performance with previous methods across a variety of datasets. This includes three popular CIL datasets (Rebuffi et al., 2017; Wang et al., 2022e;d; Smith et al., 2023): CIFAR100 (Krizhevsky et al., 2009), ImageNet-R (Deng et al., 2009), TinyImageNet (Le & Yang, 2015), CUB200 (Wah et al., 2011), and DomainNet (Peng et al., 2019) as well as two widely used datasets used for prompt tuning research (Zhou et al., 2022c; Yao et al., 2023; Khattak et al., 2023a;b): FGVCAircraft (Maji et al., 2013) and Stanford Cars (Krause et al., 2013). Each dataset is split into 10 tasks to formulate a CIL setup. We hereby provide a detailed description of these datasets.

**CIFAR100** consists of 60,000 32×32 color images in 100 classes, with 600 images per class. There are 500 training images and 100 testing images per class. Given that classes across different tasks may belong to the same superclass, there is a potential overlap in similarities between tasks.

**ImageNet-R** is a diverse dataset featuring renditions of ImageNet classes across various artistic and real-world domains. It includes images from categories such as art, cartoons, deviantart, graffiti, embroidery, graphics, origami, paintings, patterns, plastic objects, plush objects, sculptures, sketches, tattoos, toys, and video games. The dataset encompasses 200 classes, comprising a total of 24,000 training images and 6,000 testing images.

**TinyImageNet** is a downsampled subset of ImageNet containing 200 classes, each resized to $64 \times 64$ pixels. It is widely used as a challenging benchmark for evaluating continual learning algorithms due to its larger class set and greater intra-class variability compared to CIFAR100.

**CUB200** is designed for fine-grained visual categorization. It contains 11,788 images of 200 bird species, each with detailed annotations. This dataset is particularly challenging due to the subtle differences between species.

**FGVCAircraft** is a specialized dataset for fine-grained visual categorization, focusing on aircraft models. It includes a variety of images depicting different aircraft models, presenting challenges in model identification. We randomly select 100 classes from this dataset.

**Stanford Cars** contains a collection of 16,185 images spread across 196 car classes. Each image is annotated with details such as the make, model, and year of the car. This dataset is primarily used for fine-grained car recognition tasks, especially useful for distinguishing between visually similar car models. Similar to FGVCAircraft, we use a random sample of 100 classes for model training and testing.

**DomainNet** is a large-scale multi-domain image dataset with roughly 0.6M images spanning 345 categories across six visual domains (clipart, infograph, painting, quickdraw, real, sketch). It is widely used for domain adaptation and continual learning benchmarks to assess scalability across many classes and heterogeneous domains.

### B.2 Compared Methods

In our experiments, we conduct a comprehensive comparison of our methods against a variety of previous methods. Firstly, we assess the zero-shot performance of the pre-trained CLIP model (ZS-CLIP) (Radford et al., 2021). This is followed by comparisons with two widely recognized prompt tuning methods: CoOp (Zhou et al., 2022c) and VPT-CL (our implementation of VPT (Jia et al., 2022)), adapted to the CIL scenario. Further, we compare against several popular CL methods: L2P (Wang et al., 2022e), DualPrompt (Wang et al., 2022d), CODA-P (Smith et al., 2023), AttriCLIP (Wang et al., 2023), ZSCL (Zheng et al., 2023), MoE-Adapter (Yu et al., 2024), InfLoRA (Liang & Li, 2024), SPU (Zhang et al., 2024), DIKI (Tang et al., 2024), CPP (Li et al., 2024), CLAP4CLIP (Jha et al., 2024) and PROOF (Zhou et al., 2023c), to provide a comprehensive evaluation of our methods.

**ZS-CLIP.** We use the pre-trained frozen image encoder and text encoder to obtain visual features and textual features, and use textual features as a classifier (Equation 1) to make predictions. Specifically, textual features are obtained through hand-crafted prompt templates as suggested by OpenAI [2] with class names.

---

[2] https://github.com/openai/CLIP

**CoOp.** CoOp introduces a method of prepending learnable prompt tokens to the text encoder of the CLIP model. This strategy is designed to adapt the pre-trained model for various downstream tasks. In our experiments, we adapt the CoOp design to fit the CIL scenario, which continually trains the textual prompts over a sequence of tasks.

**VPT-CL.** VPT is a popular prompt tuning method designed specifically for the Vision Transformer (ViT) model. In our study, we adapt VPT for use with the pre-trained CLIP model in a CIL context, and we name it VPT-CL. Specifically, we discard the classification head of VPT and make predictions using textual features as in ZS-CLIP, and we adapt this implementation to the CIL scenario that continually trains the visual prompts.

**L2P.** L2P innovates by introducing pools of prompts specifically for ViT-based Class-Incremental Learning. To remain consistent with their approach, we exclude the text encoder of the pre-trained CLIP model in our implementation.

**DualPrompt.** The DualPrompt method integrates both global and task-specific visual prompts for CIL applications. In line with the original design, the text encoder from the CLIP model is not utilized in our adaptation.

**CODA-P.** CODA-P presents a novel approach by decomposing visual prompts into individual prompt components and learning instance-wise prompts. Consistent with their methodology, we also omit the text encoder from the CLIP model in our implementation.

**AttriCLIP.** AttriCLIP enhances the textual branch of the CLIP model through the learning of text attribute prompts. These prompts are chosen from the proposed attribute bank to encode broader knowledge of textual information.

**ZSCL.** ZSCL tackles continual learning by fully fine-tuning the CLIP model on sequential tasks. To mitigate forgetting, it introduces an auxiliary reference dataset and distills knowledge from the pre-trained model back to the fine-tuned model, preserving prior knowledge.

**MoE-Adapter.** MoE-Adapter integrates a mixture-of-experts (MoE) mechanism into adapter layers to enhance adaptability in continual learning. A router dynamically selects multiple experts per layer based on inputs, enabling efficient adaptation while mitigating catastrophic forgetting.

**InfLoRA.** InfLoRA extends Low-Rank Adaptation (LoRA) for continual learning by incrementally introducing new LoRA modules per task while keeping the base model frozen. By ensuring the orthogonality of LoRA updates, it efficiently accommodates new tasks while mitigating catastrophic forgetting.

**SPU.** SPU updates a small, selected subset of backbone weights per task (typically in MLP layers) while freezing the rest. A sparsity mask (*e.g.*, top-k by importance/gradients) identifies the parameters to update, limiting growth and preserving pre-trained knowledge; gradients are computed for full layers, but only the selected weights are updated.

**DIKI.** DIKI is a parameter-efficient CL method for CLIP that adds lightweight residual modules and uses domain-aware calibration/regularization to reduce interference and retain pre-trained knowledge.

**CPP.** CPP maintains and steers image class mean prototypes with prompt tuning, and uses query–key routing to select task-specific prompts at inference, operating entirely in the visual space.

**CLAP4CLIP.** CLAP4CLIP proposes probabilistic fine-tuning of CLIP with adapters plus mixture/logit calibration across tasks; improves stability via uncertainty-aware combination of heads.

**PROOF.** Different from recent trends that mainly learn prompt tokens, PROOF focuses on learning projection layers. This method involves fusing the projections of visual and textual features and dynamically expanding these projections as new tasks are introduced during training.

Table 5: Forgetting measurements (lower is better) of different methods. All methods are trained with identical exemplar sizes, utilizing the same pre-trained weights, and employing the CLIP model with a ViT-B/16 backbone. The best two forgetting measurements are marked in **bold**.

| Methods | Forgetting | | |
|---|---|---|---|
| | CUB | Aircraft | Cars |
| CoOp | 18.47 | **14.01** | 4.76 |
| VPT | 18.08 | 32.73 | 3.78 |
| CODA-P | 18.67 | 31.44 | 5.38 |
| PROOF | 14.38 | 26.23 | 5.78 |
| TPPT-V (Ours) | **13.64** | 27.61 | **3.28** |
| TPPT-VT (Ours) | **13.11** | **25.17** | 3.34 |

Table 6: Average and final accuracy of different methods with different buffer sizes. All methods are trained with the same pre-trained weights and employ the CLIP model with a ViT-B/16 backbone. $\bar{\mathcal{A}}$ denotes the average of test accuracy over incremental stages, $\mathcal{A}_T$ denotes the test accuracy at the last incremental stage. The **best** and **secondary** performances are indicated in **red** and **blue**, respectively.

| Methods | Buffer size | CIFAR100 | | ImageNet-R | | CUB | | Aircraft | | Cars | |
|---|---|---|---|---|---|---|---|---|---|---|---|
| | | $\bar{\mathcal{A}}$ | $\mathcal{A}_T$ | $\bar{\mathcal{A}}$ | $\mathcal{A}_T$ | $\bar{\mathcal{A}}$ | $\mathcal{A}_T$ | $\bar{\mathcal{A}}$ | $\mathcal{A}_T$ | $\bar{\mathcal{A}}$ | $\mathcal{A}_T$ |
| CoOp | | 41.80 | 33.67 | 48.80 | 43.40 | 49.65 | 43.72 | 12.70 | 11.49 | 45.48 | 42.37 |
| VPT-CL | | 65.63 | 51.96 | 67.25 | 69.53 | 73.70 | 64.38 | 33.60 | 21.84 | 90.59 | 87.58 |
| CODA-P | 5/class | 74.20 | 54.63 | 60.35 | 46.28 | 70.37 | 57.34 | 28.48 | 21.21 | 89.46 | 85.05 |
| PROOF | | 73.55 | 53.00 | 74.38 | 66.40 | 74.62 | 64.84 | 44.11 | 36.45 | 88.89 | 83.46 |
| TPPT-V (Ours) | | 76.51 | 62.91 | 79.84 | 76.50 | 77.48 | 68.24 | 38.10 | 29.91 | 94.20 | 91.90 |
| TPPT-VT (Ours) | | 78.74 | 70.18 | 74.78 | 70.28 | 79.29 | 70.65 | 41.98 | 31.94 | 90.83 | 90.86 |
| CoOp | | 54.38 | 53.20 | 56.61 | 53.95 | 60.98 | 55.89 | 20.97 | 19.14 | 67.48 | 62.09 |
| VPT-CL | | 72.70 | 62.74 | 80.25 | 76.92 | 77.35 | 67.18 | 38.55 | 26.19 | 92.02 | 89.53 |
| CODA-P | 10/class | 79.90 | 65.66 | 68.41 | 56.48 | 77.54 | 67.47 | 36.90 | 33.72 | 93.33 | 89.79 |
| PROOF | | 76.23 | 65.17 | 74.90 | 67.70 | 74.73 | 65.39 | 44.83 | 37.14 | 89.15 | 83.22 |
| TPPT-V (Ours) | | 82.41 | 72.42 | 83.24 | 78.87 | 82.48 | 74.72 | 49.84 | 42.78 | 94.77 | 92.77 |
| TPPT-VT (Ours) | | 78.53 | 70.20 | 79.45 | 74.70 | 82.17 | 75.49 | 53.04 | 45.03 | 95.90 | 94.49 |
| CoOp | | 60.33 | 54.30 | 61.61 | 57.07 | 63.59 | 58.58 | 24.50 | 23.61 | 45.15 | 76.67 |
| VPT-CL | | 76.92 | 67.06 | 69.42 | 76.40 | 79.18 | 69.40 | 40.64 | 28.53 | 93.06 | 89.73 |
| CODA-P | 15/class | 82.78 | 70.12 | 72.50 | 61.50 | 79.69 | 69.55 | 44.52 | 42.75 | 94.59 | 92.07 |
| PROOF | | 81.77 | 72.58 | 75.40 | 69.18 | 75.26 | 65.52 | 45.25 | 37.59 | 89.37 | 83.70 |
| TPPT-V (Ours) | | 84.85 | 76.68 | 85.28 | 79.17 | 83.81 | 76.12 | 52.13 | 45.72 | 96.53 | 95.12 |
| TPPT-VT (Ours) | | 82.21 | 75.27 | 82.01 | 76.62 | 84.04 | 76.59 | 57.95 | 50.50 | 95.70 | 94.11 |
| CoOp | | 83.37 | 73.36 | 82.40 | 76.20 | 77.34 | 68.70 | 44.26 | 39.87 | 89.73 | 84.91 |
| VPT-CL | | 87.11 | 78.21 | 84.60 | 78.97 | 80.81 | 69.89 | 35.52 | 25.50 | 93.71 | 91.49 |
| CODA-P | 20/class | 83.53 | 72.20 | 74.45 | 65.88 | 83.36 | 76.12 | 54.51 | 45.06 | 96.16 | 92.17 |
| PROOF | | 86.70 | 79.05 | 85.34 | 80.10 | 84.93 | 79.43 | 61.00 | 53.59 | 93.26 | 89.84 |
| TPPT-V (Ours) | | 88.28 | 80.81 | 85.62 | 80.75 | 85.77 | 80.28 | 63.47 | 54.85 | 95.26 | 94.57 |
| TPPT-VT (Ours) | | 87.20 | 78.98 | 84.87 | 81.02 | 86.60 | 81.55 | 69.34 | 61.99 | 96.83 | 95.37 |

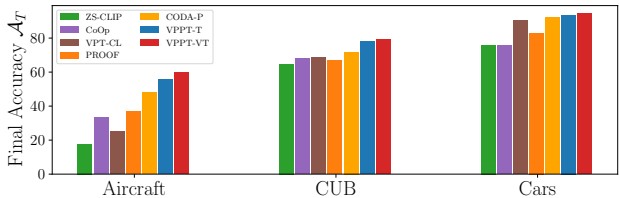

Figure 10: Experiment results of different methods on 20-task split incremental setting.

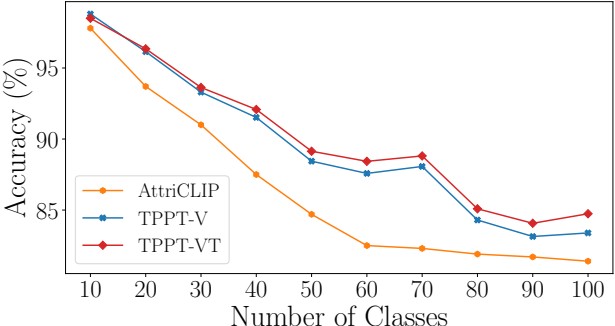

Figure 11: Comparisons to AttriCLIP on CIFAR-100 using ViT-L/14 with pre-trained weight from OpenAI(Radford et al., 2021).

## C   More Experimental Results and Ablation Studies

**Additional comparison with AttriCLIP (Wang et al., 2023).** For a fair comparison, we compare our methods with AttriCLIP reported results under the same experiment settings (Wang et al., 2023) on CIFAR-100 in Fig. 11. The backbone model employed was CLIP with a ViT-L/14 architecture, utilizing pre-trained weights from OpenAI. Throughout the incremental learning process, our approach demonstrated superior performance, consistently outperforming AttriCLIP.

**Additional comparison with SPU (Zhang et al., 2024).** In Table 10, we reproduce SPU under our experimental setup, using the same memory buffer construction method (iCaRL, consistent with our baseline implementations), task splits, and training budget as in the main experiments. Unlike TPPT, which applies parameter-efficient (PEFT) soft prompts, SPU performs sparse parameter updates directly on the MLP layer, which requires gradient computation over the entire linear layer rather than only a few trainable prompt tokens as in TPPT. Consequently, SPU can potentially allow more adaptation than PEFT modules, as reflected in its higher performance on CIFAR100 (a simpler, low-resolution dataset). On fine-grained datasets with more diverse and complex representations, however, TPPT (especially TPPT-VT) shows significant gains over SPU, suggesting that learning with textual anchors is more effective for continual learning of complex semantics. Moreover, regularized textual prototypes further boost performance to state-of-the-art levels (notably on Aircraft and Cars).

**Comparisons on larger-scale DomainNet (Peng et al., 2019).** We evaluate on DomainNet with 345 classes, splitting the label set into sequences of 5, 15, 30, and 69 tasks. As shown in Table 11, TPPT consistently outperforms prior prompt-based continual learning methods across all task sequence lengths,

Table 7: **Upper-bound** performance of our methods.

|          | CIFAR100 | ImageNet-R | CUB   | Aircraft | Cars  |
| -------- | -------- | ---------- | ----- | -------- | ----- |
| TPPT-V   | 87.29    | 85.05      | 79.64 | 59.35    | 95.88 |
| TPPT-VT  | 86.93    | 85.23      | 79.11 | 63.92    | 96.02 |

Table 8: Final Accuracies over 5 independent runs with varying order of seen classes.

| Methods | CIFAR100 | ImageNet-R | CUB | Aircraft | Cars |
|---|---|---|---|---|---|
| TPPT-V | $79.57 \pm 0.64$ | $80.65 \pm 0.29$ | $79.98 \pm 0.38$ | $55.22 \pm 0.66$ | $95.25 \pm 0.81$ |
| TPPT-VT | $78.11 \pm 0.34$ | $80.64 \pm 0.67$ | $80.33 \pm 0.82$ | $61.52 \pm 0.53$ | $94.61 \pm 0.59$ |

Table 9: Average and final accuracy of different methods using **OpenAI** (Radford et al., 2021) pre-trained weight. All methods are trained with identical exemplar sizes, utilizing the same pre-trained weights, and employing the CLIP model with a ViT-B/16 backbone. $\bar{\mathcal{A}}$ denotes the average of test accuracy over incremental stages, $\mathcal{A}_T$ denotes the test accuracy at the last incremental stage. The **best** and **secondary** performances are indicated in **red** and **blue**, respectively.

| Methods | CIFAR100 | | ImageNet-R | | CUB | | Aircraft | | Cars | |
|---|---|---|---|---|---|---|---|---|---|---|
| | $\bar{\mathcal{A}}$ | $\mathcal{A}_T$ | $\bar{\mathcal{A}}$ | $\mathcal{A}_T$ | $\bar{\mathcal{A}}$ | $\mathcal{A}_T$ | $\bar{\mathcal{A}}$ | $\mathcal{A}_T$ | $\bar{\mathcal{A}}$ | $\mathcal{A}_T$ |
| ZS-CLIP | 74.47 | 65.92 | 79.34 | 72.37 | 74.77 | 64.67 | 33.71 | 23.01 | 83.78 | 76.11 |
| CoOp | 81.27 | 70.30 | 81.91 | 76.23 | 76.22 | 68.21 | 46.09 | 38.87 | 81.12 | 74.87 |
| VPT-CL | 83.95 | 75.74 | 85.27 | 79.93 | 77.82 | 67.22 | 50.39 | 38.25 | 88.31 | 82.37 |
| L2P | 79.35 | 69.82 | 76.84 | 68.95 | 77.00 | 68.16 | 55.21 | 43.79 | 82.49 | 77.48 |
| DualPrompt | 81.49 | 74.69 | 81.49 | 75.03 | 82.15 | 75.93 | 55.61 | 44.72 | 82.99 | 78.07 |
| CODA-P | 83.88 | 73.63 | 76.05 | 65.65 | 80.59 | 71.16 | 47.50 | 48.66 | 90.81 | 86.78 |
| PROOF | **84.97** | 77.30 | 83.66 | 78.31 | 83.83 | **78.28** | 57.09 | 50.44 | 88.29 | 82.66 |
| TPPT-V (Ours) | 83.88 | **77.85** | **86.12** | **80.85** | **85.32** | 77.01 | **62.50** | **52.75** | **92.06** | **87.97** |
| TPPT-VT (Ours) | **85.72** | **77.75** | **85.90** | **80.10** | **84.81** | **78.33** | **65.17** | **61.75** | **93.28** | **90.57** |

and the relative advantage is maintained as the number of tasks increases. Average and final accuracies remain stable rather than degrading with longer sequences, indicating that the approach scales to larger class numbers and longer task sequences.

**Discussions on pre-trained knowledge retention.** The retention of pre-trained knowledge during and after continual learning has recently attracted growing attention (Zheng et al., 2023). Following prior work (Zheng et al., 2023), we assess general knowledge by measuring zero-shot accuracy on unseen tasks and report these results across the CIFAR-100 continual learning sequence (Table 12). Results show: (a) As continual learning progresses, unseen-task accuracy generally increases because the number of unseen tasks decreases, making classification easier as the number of classes shrinks. (b) For CODA-P, a prompt-tuning CL method related to our incremental prompt framework, zero-shot performance on unseen tasks drops significantly relative to ZS-CLIP (frozen CLIP), indicating severe degradation of general knowledge. (c) For TPPT-V, our static textual-prototype anchors guide CLIP's representation learning with learnable visual prompts, keeping visual representations clustered around pretrained text prototypes and thus preserving the integrity of the embedding space. We even observe a slight improvement over frozen CLIP, suggesting that visual prompts learned from earlier tasks help the foundation model generalize to unseen tasks. (d) For TPPT-VT, we

Table 10: Comparisons with SPU (Zhang et al., 2024).

| Methods | CIFAR100 | | CUB | | Aircraft | | Stanford Cars | | TinyImageNet | |
|---|---|---|---|---|---|---|---|---|---|---|
| | $\bar{\mathcal{A}}$ | $\mathcal{A}_T$ | $\bar{\mathcal{A}}$ | $\mathcal{A}_T$ | $\bar{\mathcal{A}}$ | $\mathcal{A}_T$ | $\bar{\mathcal{A}}$ | $\mathcal{A}_T$ | $\bar{\mathcal{A}}$ | $\mathcal{A}_T$ |
| ZS-CLIP | 74.47 | 65.92 | 74.77 | 64.67 | 33.71 | 23.01 | 83.78 | 76.11 | 69.55 | 65.59 |
| SPU | **86.22** | **81.31** | 80.26 | 71.98 | 52.40 | 42.04 | 88.12 | 82.22 | 79.68 | 74.89 |
| TPPT-V (Ours) | 83.88 | **77.85** | **85.32** | **77.01** | **62.50** | **52.75** | **92.06** | **87.97** | **83.18** | **77.51** |
| TPPT-VT (Ours) | **85.72** | 77.75 | **84.81** | **78.33** | **65.17** | **61.75** | **93.28** | **90.57** | **82.41** | **78.08** |

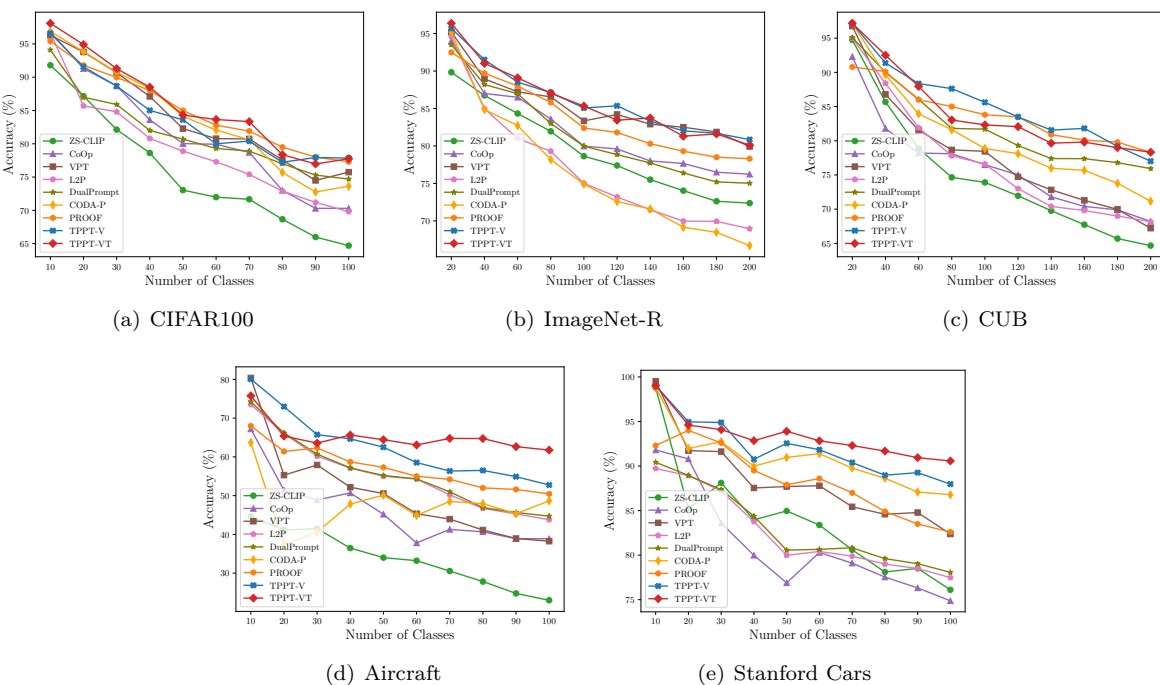

Figure 12: Performance of different methods across incremental stages using pre-trained weights from OpenAI. All methods are trained with the same exemplar size using the same pre-trained weight and backbone model of CLIP with ViT-B/16.

Table 11: Comparisons on DomainNet.

| Methods | DomainNet | | | | | | | |
| | 5 Tasks | | 15 Tasks | | 30 Tasks | | 69 Tasks | |
| | $\bar{\mathcal{A}}$ | $\mathcal{A}_T$ | $\bar{\mathcal{A}}$ | $\mathcal{A}_T$ | $\bar{\mathcal{A}}$ | $\mathcal{A}_T$ | $\bar{\mathcal{A}}$ | $\mathcal{A}_T$ |
|---|---|---|---|---|---|---|---|---|
| ZS-CLIP | 63.94 | 56.66 | 63.31 | 56.66 | 63.66 | 56.66 | 63.28 | 56.66 |
| CoOp | 57.45 | 46.20 | 55.97 | 49.24 | 48.45 | 47.96 | 46.38 | 50.07 |
| VPT-CL | 68.70 | 59.42 | 70.42 | 62.78 | 68.74 | 61.24 | 71.84 | 62.49 |
| CODA-P | 68.04 | 59.09 | 70.68 | 61.94 | 69.72 | 61.75 | 70.45 | 61.25 |
| TPPT-V | **72.70** | **63.73** | **73.28** | **63.80** | **72.81** | **63.20** | **74.24** | **64.69** |
| TPPT-VT | **73.70** | **65.34** | **73.19** | **64.08** | **72.81** | **62.90** | **73.78** | **63.63** |

unfreeze textual prototypes with learnable text prompts to capture more representative linguistic semantics beyond pretrained features. This initially reduces unseen accuracy relative to TPPT-V (especially in the first few tasks) because the text embedding space is being adjusted to learn more representative prototypes. Notably, after Task 6, unseen accuracy slightly surpasses ZS-CLIP, indicating that TPPT-VT benefits from learning generalizable multimodal semantics over a few continual learning stages.

**Experiment results on Upper-bound performance.** In complement to the incremental learning performance in Table 1 and Fig. 4, we further report the Upper-bound performance of our methods in Table 7, measuring the learning capacity of the model under a non-incremental setting. Notably, for fine-grained datasets such as CUB, Aircraft, and Cars, the performance gap between the final accuracy reported in Table 1 and the Upper-bound performance is marginal, highlighting the effectiveness of the introduction of textual prototypes, which greatly stabilizes the training during the incremental phases.

Table 12: Unseen-task zero-shot accuracy across tasks. Deltas are w.r.t. ZS-CLIP at each step (+ improvement, – degradation).

| Methods | Tasks Seen | | | | | | | | |
|---|---|---|---|---|---|---|---|---|---|
| | 1 | 2 | 3 | 4 | 5 | 6 | 7 | 8 | 9 |
| ZS-CLIP | 66.61 | 68.14 | 68.41 | 69.12 | 71.44 | 73.85 | 75.90 | 80.15 | 85.00 |
| CODA-P | 28.30 (-38.31) | 41.10 (-27.04) | 47.33 (-21.08) | 47.90 (-21.22) | 52.30 (-19.14) | 55.80 (-18.05) | 62.00 (-13.90) | 70.50 (-9.65) | 82.60 (-2.40) |
| TPPT-V | 69.49 (+2.88) | 72.58 (+4.44) | 73.66 (+5.25) | 72.57 (+3.45) | 73.44 (+2.00) | 77.65 (+3.80) | 80.47 (+4.57) | 82.35 (+2.20) | 87.60 (+2.60) |
| TPPT-VT | 45.72 (-20.89) | 58.08 (-10.06) | 64.44 (-3.97) | 69.90 (+0.78) | 69.88 (-1.56) | 75.45 (+1.60) | 78.70 (+2.80) | 81.35 (+1.20) | 90.40 (+5.40) |

**Studies on Forgetting.** In Table 1, we present the average and final accuracies of our proposed methods, alongside comparisons with previous works. Additionally, we evaluate the average forgetting of our methods in Table 5 following (Wang et al., 2022e;d; Smith et al., 2023). We formally denote the forgetting metric for a given task as: $\mathcal{F}_t = \frac{1}{t-1}\sum_{\tau=1}^{t-1}\max_{\tau'\in\{1,\cdots,t-1\}}(\mathcal{A}_{\tau',\tau} - \mathcal{A}_{t,\tau})$, where $\mathcal{A}_{t,\tau}$ represents the accuracy on task $\tau$ after training on task $t$. We then evaluate average of forgetting over tasks $\tilde{\mathcal{F}} = \frac{1}{T}\sum_{t=1}^{T}\mathcal{F}_t$. The results demonstrate that our methods not only consistently outperform the previous approaches but also exhibit low levels of forgetting, thus effectively preventing catastrophic forgetting. Benefited from our design of learning with textual prototypes, our method is inherently less prone to forgetting. Enriching textual prototypes with textual diversity not only enhances the stability and efficacy of the training process but also slightly improves the forgetting measure. While CoOp achieves lower forgetting on the Aircraft dataset, it is important to note the substantial difference in prediction accuracy favoring our methods. Given this considerable performance gap, our proposed methods stand out as superior in terms of overall efficacy.

**Analyses on reply buffer.** To provide a comprehensive understanding, we explore the impact of different buffer sizes on performance. These additional evaluations, including comparisons with previous works, are detailed in Table 6. The results from these experiments indicate that our proposed methods consistently outperform or are comparable to previous methods, demonstrating robustness irrespective of the chosen buffer size. A notable aspect of our methods is the effectiveness even with a limited number of exemplars. For instance, when using only 5 exemplars per task, TPPT-VT achieves an average accuracy of 79.29 on the CUB dataset. This performance is on par with that of previous works using a larger number of exemplars, which attain an average accuracy of 79.69 using 15 exemplars per task. Such results show the efficiency and competitiveness of our methods even under constrained exemplar conditions.

**Experiment results with a larger number of tasks.** Evaluation results in Table 1 and Fig. 4 were conducted under our default experimental setup, where each dataset is divided into 10 tasks for incremental learning. In Fig. 10, we extend our evaluation to a more challenging experiment setting by splitting each dataset into 20 tasks, which increases the complexity of the learning process and makes the model more prone to forgetting. Despite the increased complexity, our methods, which leverage learning with textual prototypes and the diversity regularization of these prototypes, consistently outperform previous methods in the 20-task split setting.

**Experiment results with varying class orders.** In Table 8, we perform 5 independent runs and report the average final accuracy and standard deviation. Our method is robust to different class orderings, showing its robust performance across varied learning scenarios.

**Computational efficiency of our methods.** In addition to the discussion of our parameters in Sec. 4.4, we hereby discuss the computation efficiency of our methods. In Table 13, we report per-image training/inference time, peak GPU memory, and estimated FLOPs (GFLOPs/image) under identical hardware, batch size, and data pipelines, comparing TPPT with related continual prompt-tuning methods. All experiments are run using one NVIDIA RTX3090 GPU. CODA-P uses a similar design of incremental prompts to us, takes approximately 8ms for training and 5ms for inference per image. TPPT-V takes a comparable or less amount of time, as it does not require additional learnable attention vectors, yet it delivers a significant performance boost of 11% in final accuracy averaged across all datasets compared to CODA-P. TPPT-VT additionally learns textual prompts and further increases final accuracy on the Aircraft dataset by 13%, with only an additional 1.5ms required for both training and inference compared to TPPT-V.

Table 13: Detailed computation costs. We report per-image train/inference time (ms) and peak GPU memory usage (MiB), and estimated FLOPS (GFLOPs/image).

| Methods | Train | | Inference | | FLOPs |
|---|---|---|---|---|---|
| | Time | Memory | Time | Memory | |
| ZS-CLIP | – | – | 2.9 | 1210.17 | 33.71 |
| CoOp | 3.8 | 2171.6 | 3.2 | 1226.4 | 38.7 |
| VPT | 6.2 | 5928.0 | 2.9 | 1161.6 | 37.1 |
| L2P | 6.9 | 5963.3 | 2.9 | 1472.2 | 68.3 |
| DualPrompt | 7.9 | 5775.1 | 2.9 | 1578.4 | 68.8 |
| CODA-P | 8.2 | 7974.3 | 5.4 | 1653.7 | 68.1 |
| TPPT-V (Ours) | 8.2 | 7518.1 | 5.4 | 1335.6 | 68.1 |
| TPPT-VT (Ours) | 9.7 | 8821.0 | 5.9 | 1344.1 | 72.7 |

Table 14: Experiment results on SigLIP (Zhai et al., 2023).

| Methods | CIFAR100 | | CUB | | Aircraft | |
|---|---|---|---|---|---|---|
| | $\bar{\mathcal{A}}$ | $\mathcal{A}_T$ | $\bar{\mathcal{A}}$ | $\mathcal{A}_T$ | $\bar{\mathcal{A}}$ | $\mathcal{A}_T$ |
| ZeroShot | 79.07 | 70.99 | 78.71 | 67.43 | 51.56 | 40.74 |
| CoOp | 77.97 | 69.44 | 75.97 | 64.29 | 48.06 | 39.63 |
| VPT | 85.99 | 77.90 | 80.23 | 69.80 | 51.51 | 46.28 |
| CODA-P | 86.69 | 79.42 | 86.11 | 80.36 | 71.33 | 69.43 |
| TPPT-V (Ours) | **89.16** | **81.71** | **88.18** | **82.19** | **72.00** | **69.91** |
| TPPT-VT (Ours) | **88.99** | **81.64** | **87.90** | **82.40** | **74.81** | **71.08** |

**Experiment results using OpenAI pre-trained CLIP weight.** In our main paper, we mainly report the performance using pre-trained weights from LAION-400M (Ilharco et al., 2021). To demonstrate the robustness of our proposed methods, we further evaluate our performance using pre-trained weights from OpenAI (Radford et al., 2021) in Table 9, and we illustrate the test accuracy over incremental stages in Fig. 12. Our experiment results demonstrate that our proposed methods are insensitive to the selection of pre-trained weights, consistently outperforming previous methods.

Specifically, **TPPT-V**, which employs static textual prototypes, demonstrates performance that is comparable to or surpasses that of previous methods across all evaluated datasets. This underscores the simplicity and effectiveness of our design. Notably, **TPPT-VT** that learns textual prompts to create regularized textual prototypes with diversity, leads to superior performance across all metrics when compared to previous methods. In comparing our two proposed methods, we observe that learning textual prototypes is particularly beneficial for fine-grained datasets such as CUB, Aircraft, and Stanford Cars. This finding highlights the effectiveness of learning regularized textual prototypes. Such an approach is beneficial in preserving the maximum information of the datasets and promoting uniformity within the embedding space.

**Generalization to other VLMs.** To further validate effectiveness, we evaluate TPPT on SigLIP (Zhai et al., 2023), whose pretraining differs from CLIP and yields stronger zero-shot performance. Despite these differences, TPPT, guided by stable textual anchors, consistently improves the SigLIP backbone across diverse downstream tasks (coarse-grained CIFAR100 and fine-grained CUB and Aircraft), achieving state-of-the-art results relative to other continual prompt tuning methods.

