# OpenReview forum: "Continual Learning on CLIP via Incremental Prompt Tuning with Intrinsic Textual Anchors"
_TMLR — Accepted by TMLR_

### Review · Reviewer_vrhc · 2025-09-08

**Summary Of Contributions:**

Summary Of Contributions
The paper introduces Textual Prototype-guided Prompt Tuning (TPPT) for class-incremental learning (CIL) with CLIP. Two variants are proposed: TPPT-V, which uses fixed textual prototypes as anchors for visual prompt learning, and TPPT-VT, which additionally learns textual prompts with a diversity regularizer to avoid collapse. A new loss, Textual Prototypical Contrastive Loss (TPCL), is used to align visual prompts with textual prototypes while separating them from others. The methods are evaluated across multiple datasets, with reported gains over existing prompt-based baselines.

Key Strengths
- Clear and easy-to-follow writing.
- A simple, conceptually intuitive idea: using textual prototypes as anchors.
- Promising results, with evidence of robustness across different CLIP initializations.
- Ablation studies on drift, diversity, and parameter counts.

Key Weaknesses
- Incremental novelty: TPCL is a straightforward adaptation of prior contrastive/prototypical losses, and TPPT overlaps conceptually with prior work (e.g., DIKI [A], CPP [B]).
- Missing comparisons with several strong baselines (e.g., DIKI [A], LoRA-based methods, MoE-Adapter, CLAP4CLIP [C], CPP [B], TreeProbe [D], AnytimeCL [E], CGIL [F]).
- Claims of efficiency are not supported: no FLOP, runtime, GPU memory, or inference-time comparisons are provided. Parameter counts alone are insufficient.
- Experiments limited in scale: tested only up to ~200 classes, not on larger benchmarks like ImageNet-1k.
- Method relies on replay buffers, but this is not clearly described in the main paper.
- Technical clarity issues (e.g., missing details on aggregation weights, unclear symbols in equations, small and hard-to-read figures).

[A] Tang, Longxiang, et al. "Mind the interference: Retaining pre-trained knowledge in parameter efficient continual learning of vision-language models." ECCV2024

[B] Li, Z., et al. "Steering prototypes with prompt-tuning for rehearsal-free continual learning." WACV 2024.

[C] Saurav Jha, et al. "CLAP4CLIP: Continual learning with probabilistic finetuning for vision-language models." NeurIPS 2024

[D] Zhu, Z., et al. "Continual Learning in Open-vocabulary Classification with Complementary Memory Systems." TMLR 2024.

[E] Zhu, Z., et al.  "Anytime Continual Learning for Open Vocabulary Classification." ECCV 2024.

[F] Frascaroli, Emanuele, et al. "Clip with generative latent replay: a strong baseline for incremental learning." BMVC 2024

**Audience:**

Yes

**Audience Explanation:**

Continual learning with large vision-language models like CLIP is an active and important area. The idea of using textual prototypes as anchors is simple and interpretable, and the reported results are promising. Even if the novelty is limited, the paper would likely interest researchers working on parameter-efficient continual learning.

**Broader Impact Concerns:**

No significant ethical concerns were identified. The method introduces no obvious risks beyond those typically associated with continual learning research.

**Claims And Evidence:**

No

**Claims Explanation:**

The paper provides empirical improvements over prompt-based baselines and includes ablation studies that support some of the claims. However, the absence of critical runtime/efficiency analysis, limited benchmark coverage, and missing comparisons with stronger recent methods weaken the evidence. Furthermore, some design choices (e.g., handling of forgetting, scalability of objectives) are not sufficiently justified.

**Requested Changes:**

*Critical*:
- Include comparisons with closely related methods (e.g., DIKI, CPP, LoRA-based continual learning, CLAP4CLIP, MoE-Adapter, TreeProbe, AnytimeCL, CGIL).
- Provide efficiency analyses beyond parameter counts: report training time, inference time, FLOPs, and GPU memory usage.
- Clarify novelty relative to prior prototype-anchored prompt tuning methods (CPP, prototypical contrastive loss).
- Provide more experimental evidence on larger-scale datasets (e.g., ImageNet-1k or MCIL benchmarks).

*Strengthening* (not critical but recommended):
- Improve clarity of exposition: define all terms and variables (e.g., N in Eq. 3), expand details on aggregation weights, and make figures/tables more readable.
- Explicitly describe the reliance on replay buffers in the main paper, not just the supplementary materials.
- Discuss the scalability of TPPT-VT with respect to memory growth when the number of classes becomes large.
- Reposition title/claims more narrowly around CIL, since other continual learning settings are not explored.

---

> ### Author Response · Authors · 2025-11-06
> **Rebuttal (1/3)**
>
> We sincerely thank you for the insightful and detailed feedback. We appreciate your recognition of the clarity of our motivation and overall soundness of the experimental results. Your comments regarding baselines, efficiency evaluation, and broader comparisons are especially valuable, and we have carefully addressed each point in our revision to further strengthen the paper’s completeness and reproducibility.
>
> 1. **Discussions on prototypical contrastive learning.**
>
> (a) **Textual Prototypical Contrastive Loss (TPCL)**. Our main contribution is to introduce **fixed/learned textual prototypes** into the continual learning process for CLIP, fully utilizing CLIP’s intrinsic capabilities. These prototypes serve as **stable semantic anchors** that guide and stabilize visual embedding updates as new tasks arrive. In contrast to conventional contrastive or prototypical approaches that rely on **image prototypes computed as class-wise feature means from the current model**, and thus **are prone to drift when later tasks are learned**, TPCL leverages **pre-trained text embeddings** to **preserve the semantic structure** of CLIP’s embedding space and **mitigate representation drift and forgetting** during continual adaptation.
>
>
> (b) **Diversity on textual anchors.** Text-side diversity prevents collapse and reduces interference. We **exclusively** regularize textual anchors to be **uniformly distributed** on the unit hypersphere, enlarging angular margins, reducing cross-class gradient interference, and stabilizing cross-modal alignment. This improves both visual and textual manifolds and reduces correlated forgetting/drift as tasks accumulate.
>
>
> 2. **Computation cost.** As we discussed in **Sec. 4.4 and Appendix C** on  trainable-parameter counts and the compute discussion, **TPPT-V** requires **comparable or less** time because it introduces no additional learnable attention vectors, yet **delivers a +11 percentage-point gain** in final average accuracy across datasets relative to CODA-P; **TPPT-VT** additionally learns textual prompts and increases Aircraft final accuracy by **+13 percentage points**, with only **~1.5 ms** extra training time per image and **marginal increases** in peak GPU memory and FLOPs relative to TPPT-V. We here provide detailed computation costs to supplement those discussions, incluing **per-image training/inference time, peak GPU memory, and estimated FLOPs** under identical hardware, batch size, and data pipelines, comparing TPPT with related continual prompt-tuning methods.
>
>
> **Table 13 in appendix of revision.** Detailed computation costs. We report per-image train/inference time (ms) and peak GPU memory usage (MiB), and estimated FLOPS (GFLOPs/image).
>
> | Methods | Train |  | Inference |  | FLOPs |
> |---|:---:|:---:|:---:|:---:|:---:|
> | Methods | Time | GPU Memory | Time | GPU Memory |  |
> | ZS-CLIP |  |  | 2.9 | 1210.17 | 33.71 |
> | CoOp | 3.8 | 2171.6 | 3.2 | 1226.4 | 38.7 |
> | VPT | 6.2 | 5928.0 | 2.9 | 1161.6 | 37.1 |
> | L2P | 6.9 | 5963.3 | 2.9 | 1472.2 | 68.3 |
> | DualPrompt | 7.9 | 5775.1 | 2.9 | 1578.4 | 68.8 |
> | CODA-P | 8.2 | 7974.3 | 5.4 | 1653.7 | 68.1 |
> | TPPT-V (Ours) | 8.2 | 7518.1 | 5.4 | 1335.6 | 68.1 |
> | TPPT-VT (Ours) | 9.7 | 8821.0 | 5.9 | 1344.1 | 72.7 |

---

> > ### Author Response · Authors · 2025-11-06
> > **Rebuttal (2/3)**
> >
> > 3. **Extended comparisons and discussions.** Thank you for the suggestions! We reproduced several methods under our setting (see revised Table 2), using the identical memory construction (iCaRL, consistent with our baselines), task splits, and training budget. Experiments show that TPPT consistently outperforms these methods; below we clarify conceptual differences and the significance of TPPT, and we have updated Sec. 4.2 and Appendix A.2 accordingly.
> >
> > (a) **DIKI [A], CPP [B], LoRA-based continual learning, CLAP4CLIP [C], MoE-Adapter.** (1) Compared to **DIKI**, which retains pre-trained knowledge via domain-aware calibration but lacks text-anchored supervision and thus can **still exhibit representation drift**, TPPT brings CLIP text-encoder prototypes (both fixed and learnable via text prompts) into the learning as **stable semantic anchors**, and applies TPCL (plus text-side diversity) to align new visual features to these anchors while repelling others, **stabilizing the image–text embedding space** and mitigating correlated forgetting. (2) Compared to **CPP**, which steers **image prototypes** with task-specific prompts and relies on prompt routing at inference, but it **lacks cross-modal stabilization** because supervision remains intra-visual, thus **are prone to representation drift when later tasks are learned**. TPPT aligns visuals to textual prototypes and regularizes anchor diversity, reducing drift and prototype collapse.  (3) Compared to **CLAP4CLIP**, which uses probabilistic fine-tuning with adapters and mixtures of task logits, it can still leave the **image–text embedding space under-constrained**. TPPT uses a prompt-only, text-anchored contrastive objective with textual-anchor diversity to stabilize the embedding space and mitigate forgetting with lower overhead. (4) As shown in **Table 2**, TPPT consistently outperforms LoRA-based continual learning (**InfLoRA**) and **MoE-Adapter**.
> >
> > (b) **TreeProbe [D], AnytimeCL [E], CGIL [F].** These works primarily target **buffer quality/efficiency** or **generative replay**, which is **out of scope** in our study that focuses on representation learning. (1) **TreeProbe** combines a fixed CLIP zero-shot head with an exemplar model and **gates memory usage**. We do not redesign memory gating and instead improve representation learning via textual anchors. (2) **AnytimeCL** emphasizes update scheduling and memory compression (e.g., attention-weighted PCA); our scope is what to learn: text-anchored prompt tuning to stabilize features, rather than when/how to update or compress memory. (3) **CGIL** uses generative latent replay (VAE) to populate the buffer with synthetic examples; TPPT does not introduce a new replay mechanism and remains focused on textual-anchor–guided learning, with advanced replay methods out of scope for this work.
> >
> > **Updated entries of Table 2 in revision**
> >
> > | Methods | CIFAR100 |  | TinyImageNet |  |
> > |---|:---:|:---:|:---:|:---:|
> > |  | $\bar{\mathcal{A}}$ | $\mathcal{A}_T$ | $\bar{\mathcal{A}}$ | $\mathcal{A}_T$ |
> > | DIKI | 84.43 | 76.14 | 81.29 | 75.66 |
> > | CPP | 83.65 | 75.57 | 77.92 | 72.22 |
> > | CLAP4CLIP | 82.09 | 76.77 | 80.9 | 74.44 |
> > | TPPT-V | 83.88 | **77.85** | **83.18** | **77.51** |
> > | TPPT-VT | **85.72** | **77.75** | **82.41** | **78.08** |
> >
> > [A] Tang, Longxiang, et al. "Mind the interference: Retaining pre-trained knowledge in parameter efficient continual learning of vision-language models." ECCV2024
> >
> > [B] Li, Z., et al. "Steering prototypes with prompt-tuning for rehearsal-free continual learning." WACV 2024.
> >
> > [C] Saurav Jha, et al. "CLAP4CLIP: Continual learning with probabilistic finetuning for vision-language models." NeurIPS 2024
> >
> > [D] Zhu, Z., et al. "Continual Learning in Open-vocabulary Classification with Complementary Memory Systems." TMLR 2024.
> >
> > [E] Zhu, Z., et al. "Anytime Continual Learning for Open Vocabulary Classification." ECCV 2024.
> >
> > [F] Frascaroli, Emanuele, et al. "Clip with generative latent replay: a strong baseline for incremental learning." BMVC 2024.

---

> > > ### Author Response · Authors · 2025-11-06
> > > **Rebuttal (3/3)**
> > >
> > > 4. **Larger-scale evaluation and scalability.** Thank you for the suggestion. (a) To nevertheless assess scalability with substantially more classes and longer task sequences, we evaluate on **DomainNet** under CIL and split the **345** categories into sequences of 5, 15, 30, and **69** tasks. The results show that TPPT maintains or widens its advantage as the sequence length grows, supporting the claim that our textual-anchor training is effective at scale. (b) **Scalability with parameter growth.** Similarly as in prior works (DualPrompt, CODA-P), TPPT learns a new set of prompts per incoming task, yielding a linear parameter increase with the number of tasks (the backbone stays frozen). To curb this growth, we outline two practical directions for future work: (1) **Dynamic prompt addition**: before adding prompts, detect whether a new task’s knowledge is partially captured by existing prompts; (2) **Post-hoc prompt consolidation**: after each task, merge similar prompts by analyzing prompt similarity and the associated data distributions.
> > >
> > > **Table 11 in appendix of revision.** Comparisons on DomainNet.
> > > | Methods | DomainNet |  |  |  |  |  |  |  |
> > > |:---:|:---:|:---:|:---:|:---:|:---:|:---:|:---:|:---:|
> > > |  | 5 Tasks |  | 15 Tasks |  | 30 Tasks |  | 69 Tasks |  |
> > > |  | $\bar{\mathcal{A}}$ | $\mathcal{A}_T$ | $\bar{\mathcal{A}}$ | $\mathcal{A}_T$ | $\bar{\mathcal{A}}$ | $\mathcal{A}_T$ | $\bar{\mathcal{A}}$ | $\mathcal{A}_T$ |
> > > | ZS-CLIP | 63.94 | 56.66 | 63.31 | 56.66 | 63.66 | 56.66 | 63.28 | 56.66 |
> > > | CoOp | 57.45 | 46.20 | 55.97 | 49.24 | 48.45 | 47.96 | 46.38 | 50.07 |
> > > | VPT-CL | 68.70 | 59.42 | 70.42 | 62.78 | 68.74 | 61.24 | 71.84 | 62.49 |
> > > | CODA-P | 68.04 | 59.09 | 70.68 | 61.94 | 69.72 | 61.75 | 70.45 | 61.25 |
> > > | TPPT-V | **72.70** | **63.73** | **73.28** | **63.80** | **72.81** | **63.20** | **74.24** | **64.69** |
> > > | TPPT-VT | **73.70** | **65.34** | **73.19** | **64.08** | **72.81** | **62.90** | **73.78** | **63.63** |
> > >
> > > **Minors:**
> > > 1. **Usage of replay buffer.** Thanks for your suggestions! We have included this specification in the implementation details (Sec. 4.1) for clarity and completeness.
> > >
> > > 2. **List of revisions to improve clarity and readability:**
> > >
> > > (a) We provide additional details on prompt aggregation (Eq. 2), including how aggregation weights are computed via a lightweight, trainable linear layer applied to frozen image-encoder features.
> > >
> > > (b) We properly define the notation of $N$ in Eq. (3) as the number of training samples in the current mini-batch
> > >
> > > \(c) We have increased the font and figure sizes, where appropriate, in Figs. 2 and 5–9.
> > >
> > > (d) Thank you for your suggestion! In the revised version, we have further highlighted the focus on class-incremental learning (CIL) at the beginning of the Introduction, complementing and reinforcing the original statement at its end. We have also clarified that the scope of the paper is now explicitly narrowed to CIL. If necessary, we are open to further adjusting the paper’s title to reflect this focus.
> > >
> > >
> > >  We are grateful for your careful and constructive review. Your detailed suggestions helped sharpen the exposition and improve the overall presentation of the paper. We have updated the manuscript accordingly and highlighted all revisions with updated experimental results in blue for easy reference. Thank you again for your time and insight.

---

> ### Comment · Reviewer_vrhc · 2025-11-17
>
> Thank you to the authors for the thorough and constructive revision. The updated manuscript addresses the major concerns raised in my initial review. In particular, the inclusion of stronger baselines (DIKI, CPP, LoRA-based methods, CLAP4CLIP, MoE-Adapter), the detailed efficiency analysis (training/inference time, GPU memory, FLOPs), and the expanded comparisons and clarifications around novelty substantially strengthen the empirical and conceptual grounding of the work. The added DomainNet experiments meaningfully improve the evidence for scalability, and the revisions to clarity, notation, and replay-buffer description noticeably improve readability (there is a typo in: "We sample 20 exempaers per class as our replay buffer ... ")
>
> While some limitations remain (e.g., absence of larger-scale benchmarks such as ImageNet-1k), the authors have convincingly resolved the core issues I raised. Overall, the paper is now more complete, clearer, and better supported experimentally. I appreciate the authors’ detailed responses and revisions, and I will recommend acceptance.

---

> > ### Author Response · Authors · 2025-11-17
> >
> > We are sincerely grateful to the reviewer for your exceptionally thorough and constructive feedback, which was instrumental in strengthening our manuscript, and we deeply appreciate your recommendation for acceptance.

---

### Review · Reviewer_fq2A · 2025-09-18

**Summary Of Contributions:**

### Summary
This paper introduces TPPT (Textual Prototypical Prompt Tuning), a continual learning (CL) approach for CLIP that incrementally tunes prompts rather than the entire model. The method leverages textual prototypical contrastive loss alongside cross-entropy loss, aiming to increase diversity and reduce representational drift, thereby mitigating catastrophic forgetting. Two variants are proposed:

TPPT-V: Fixes text features and uses contrastive loss.
TPPT-VT: Extends TPPT-V by learning textual prompts and applying diversity regularization to prevent embedding collapse and improve retention.

The authors conduct comprehensive experiments on CIFAR100, ImageNet-R, etc demonstrating improvements over state-of-the-art methods,

### Strengths

- TPPT-VT excels on fine-grained continual learning benchmarks.
- Both variants are useful, with effectiveness depending on dataset characteristics.
- The method is conceptually simple yet effective, reducing the complexity of previous approaches.
- Strong ablation studies validate the contributions of the proposed losses.

### Weakness

- The comparison lacks a baseline where the CLIP model is fine-tuned in an i.i.d. setting to see the upper bound of performance on that dataset
- TPPT results for CIFAR100 differ between Table 1 and Table 2, with no explanation provided.
- The main paper omits details about the buffer size used for reported results.
- In the fine-tuning-based and adapter/LoRA-based method , sparse parameter fine-tuning methods (e.g., SPU[1]) are not compared, despite their relevance to the setting.
- The paper reports lower values for ZSCL  on CIFAR100 compared to their original paper (e.g., ZSCL’s last accuracy is 80.1).
- The paper does not address/discuss the zero shot performance degradation of CLIP’s general knowledge after continual finetuning as discussed in ZSCL[2].

[1 ]Zhang, W., Janson, P., Aljundi, R., & Elhoseiny, M. (2024). Overcoming generic knowledge loss with selective parameter update. In Proceedings of the IEEE/CVF Conference on Computer Vision and Pattern Recognition 2024.

[2] Zheng, Z., Ma, M., Wang, K., Qin, Z., Yue, X., & You, Y. (2023). Preventing zero-shot transfer degradation in continual learning of vision-language models. In Proceedings of the IEEE/CVF international conference on computer vision 2023

**Audience:**

Yes

**Audience Explanation:**

The paper addresses catastrophic forgetting in CLIP, a widely used foundation model, making it of interest to researchers in continual learning, prompt tuning, and efficient fine-tuning.

**Broader Impact Concerns:**

The broader impact statement is generally sufficient

**Claims And Evidence:**

Yes

**Claims Explanation:**

The paper effectively demonstrates the importance and relevance of the two proposed losses through controlled ablation studies. Additionally, the claims regarding representational drift and embedding diversity are clearly substantiated with supporting visualizations.

**Requested Changes:**

See the weaknesses

Minor:

- The phrase "By leveraging the powerful language modality" in the prompt selection section is misleading. The method does not directly utilize language modality for prompt selection as far as I understand; instead, it leverages powerful pre-trained features for prompt selection.

- The caption in Fig 1 (d) is incorrectly labeled as (b).

---

> ### Author Response · Authors · 2025-11-06
> **Rebuttal (1/2)**
>
> We sincerely thank you for the thoughtful and constructive feedback. We appreciate that you found our method to be well-motivated and clearly presented, and for recognizing its effectiveness and strong empirical performance in continual learning on vision-language models. Your detailed comments have been invaluable in helping us further improve the clarity, completeness, and rigor of the paper.
>
>
> 1. **Upper-bound performance**. In **Table 7** of the Appendix, we have reported the upper-bound performance obtained by fine-tuning on each dataset. Our method effectively narrows the gap between the final accuracy and this upper bound, especially on fine-grained datasets such as CUB, Aircraft, and Cars, where only a marginal difference is observed. This demonstrates the superiority of TPPT in mitigating forgetting.
>
> 2. **Details of Tables 1 and 2.** In practice, we used different pre-trained checkpoints in these two tables to ensure fair and direct comparisons with prior works. As clearly stated in the table captions, Table 1 uses CLIP weights pre-trained on LAION-400M (to align with prior methods that report results using this checkpoint), while Table 2 uses OpenAI-pretrained weights (to match methods that report results under this setting). **The choice of pre-training weights is to facilitate direct comparisons with reported results of prior works.** To further broaden the comparison of Table 1, we also include results using OpenAI weights in Table 9.
>
> 3. **Details of buffer size.** In **Appendix Table 6**, we have reported detailed results using different buffer sizes. In our main experiments, we use a buffer size of 20 by default. We have included this specification in the implementation details (Sec. 4.1) for clarity and completeness.
>
> 4. **Comparison to SPU [A].** To ensure a direct comparison, we reproduce SPU under our experimental setup, using the same memory buffer construction method (iCaRL, consistent with our baseline implementations), task splits, and training budget as in the main experiments. Unlike TPPT, which applies parameter-efficient (PEFT) soft prompts, SPU performs **sparse parameter updates** directly on the MLP layer, which requires gradient computation over the entire linear layer rather than only **a few trainable prompt tokens** as in TPPT. Consequently, SPU can potentially allow more adaptation than PEFT modules, as reflected in its higher performance on CIFAR100 (a simpler, low-resolution dataset). On **fine-grained** datasets with more diverse and complex representations, however, TPPT (especially TPPT-VT) shows **significant gains** over SPU, suggesting that learning with textual anchors is more effective for continual learning of complex semantics. Moreover, regularized textual prototypes further boost performance to state-of-the-art levels (notably on Aircraft and Cars).
>
>
> **Table 10 in appendix of revision.** Comparisons with SPU.
>
> | Methods | CIFAR100 |  | CUB |  | Aircraft |  | Stanford Cars |  | TinyImageNet |  |
> |---|:---:|:---:|:---:|:---:|:---:|:---:|:---:|:---:|:---:|:---:|
> |  | $\bar{\mathcal{A}}$ | $\mathcal{A}_T$ | $\bar{\mathcal{A}}$ | $\mathcal{A}_T$ | $\bar{\mathcal{A}}$ | $\mathcal{A}_T$ | $\bar{\mathcal{A}}$ | $\mathcal{A}_T$ | $\bar{\mathcal{A}}$ | $\mathcal{A}_T$ |
> | ZS-CLIP | 74.47 | 65.92 | 74.77 | 64.67 | 33.71 | 23.01 | 83.78 | 76.11 | 69.55 | 65.59 |
> | SPU | **86.22** | **81.31** | 80.26 | 71.98 | 52.4 | 42.04 | 88.12 | 82.22 | 79.68 | 74.89 |
> | TPPT-V (Ours) | 83.88 | **77.85** | **85.32** | **77.01** | **62.50** | **52.75** | **92.06** | **87.97** | **83.18** | **77.51** |
> | TPPT-VT (Ours) | **85.72** | 77.75 | **84.81** | **78.33** | **65.17** | **61.75** | **93.28** | **90.57** | **82.41** | **78.08** |
>
> [A] Zhang, W., et al. Overcoming generic knowledge loss with selective parameter update. CVPR 2024.
>
> 5. **Results of ZSCL.** We report the **identical** experimental results of CIFAR100 and TinyImageNet for ZSCL, as shown in **Tables 6 and 7 (10 steps experiments) of ZSCL**. The CIFAR100 result of 80.1 mentioned by the reviewer corresponds to a different MTIL experimental setting, where the **entire CIFAR100 dataset is treated as a single task**, rather than being split into 10 incremental tasks as done in our Class-Incremental-Learning experiments.

---

> > ### Author Response · Authors · 2025-11-06
> > **Rebuttal (2/2)**
> >
> > 6. **Zero-shot performance on unseen tasks.** In ZSCL, degradation of zero-shot performance is measured by their proposed **transfer** metric, computed on **zero-shot accuracy for future unseen tasks** in the continual learning sequence. We conduct a similar evaluation to assess general knowledge during continual learning by measuring **zero-shot accuracy on unseen tasks**, and we report unseen accuracy across the CIFAR-100 continual learning sequence.
> >
> > (a) As continual learning progresses, unseen-task accuracy **generally increases** because the number of unseen tasks decreases, making classification easier as the number of classes shrinks.
> >
> > (b) For **CODA-P**, a prompt-tuning CL method related to our incremental prompt framework, zero-shot performance on unseen tasks **drops significantly** relative to ZS-CLIP (frozen CLIP), indicating severe degradation of general knowledge.
> >
> > \(c) For **TPPT-V**, our **static textual prototype** anchors guide CLIP’s representation learning with **learnable visual prompts**, keeping visual representations clustered around pretrained text prototypes and thus **preserving the integrity** of the embedding space. We even observe a **slight improvement** over frozen CLIP, suggesting that visual prompts learned from earlier tasks help the foundation model generalize to unseen tasks.
> >
> > (d) For **TPPT-VT**, we **unfreeze textual prototypes** with learnable text prompts to capture **more representative linguistic semantics** beyond pretrained features. This initially reduces unseen accuracy relative to TPPT-V (especially in the first few tasks) because the text embedding space is being adjusted to learn more representative prototypes. Notably, **after Task 6**, unseen accuracy also **slightly surpasses** ZS-CLIP, indicating that TPPT-VT benefits from learning generalizable multimodal semantics over a few continual learning stages.
> >
> >
> > **Table 12 in appendix of revision.** Unseen-task zero-shot accuracy across tasks. Deltas are w.r.t. ZS-CLIP at each step.
> >
> > | **Methods** | **Tasks Seen** |  |  |  |  |  |  |  |  |
> > |---|---:|---:|---:|---:|---:|---:|---:|---:|---:|
> > |  | **1** | **2** | **3** | **4** | **5** | **6** | **7** | **8** | **9** |
> > | **ZS-CLIP** | 66.61 | 68.14 | 68.41 | 69.12 | 71.44 | 73.85 | 75.90 | 80.15 | 85.00 |
> > | **CODA-P** | 28.30 (-38.31) | 41.10 (-27.04) | 47.33 (-21.08) | 47.90 (-21.22) | 52.30 (-19.14) | 55.80 (-18.05) | 62.00 (-13.90) | 70.50 (-9.65) | 82.60 (-2.40) |
> > | **TPPT-V** | 69.49 (+2.88) | 72.58 (+4.44) | 73.66 (+5.25) | 72.57 (+3.45) | 73.44 (+2.00) | 77.65 (+3.80) | 80.47 (+4.57) | 82.35 (+2.20) | 87.60 (+2.60) |
> > | **TPPT-VT** | 45.72 (-20.89) | 58.08 (-10.06) | 64.44 (-3.97) | 69.90 (+0.78) | 69.88 (-1.56) | 75.45 (+1.60) | 78.70 (+2.80) | 81.35 (+1.20) | 90.40 (+5.40) |
> >
> > **Minors**:
> > 1. **On “leveraging the powerful language modality” in prompt selection.** Thank you for the suggestion! The prompt-selection operation uses only visual features; no textual features are involved at prompt aggregation time. Our intent was to indicate that the language branch supervises training via TPCL and the text-anchor diversity term, which indirectly shapes the learned prompts/keys and stabilizes the embedding space. To avoid ambiguity, in Sec. 3.3, we will revise the wording to: "Our concise formulation adopts the widely used query–key prompt selection/aggregation mechanism (L2P,DualPrompt,CODA-P) and simplifies prior designs by omitting incremental learnable attention vectors and orthogonal regularization (CODA-P)."
> >
> > 2. Thanks for your suggestions! We have corrected the caption labeling error in Fig. 1(d).
> >
> > We sincerely appreciate your time and thoughtful, constructive feedback, and your attention to detail has improved the clarity and overall presentation of our paper. We have revised the manuscript accordingly and marked all changes, including updated experimental results, in blue for reference.

---

> > > ### Comment · Reviewer_fq2A · 2025-11-17
> > > **Re:**
> > >
> > > Thanks authors for their rebuttal. It addressed all my concerns. Current version of the paper is more clear and have the initially missing details

---

> > > > ### Author Response · Authors · 2025-11-17
> > > >
> > > > We are delighted that we were able to address all your concerns. Thank you for acknowledging the improvements to the paper's clarity and detail.

---

### Review · Reviewer_YX6r · 2025-10-24

**Summary Of Contributions:**

This paper proposes a continual learning (CL) method for the CLIP model, named **Textual Prototype-guided Prompt Tuning (TPPT)**. The method aims to adapt CLIP incrementally to new tasks while mitigating catastrophic forgetting. The main idea is to use CLIP’s textual prototypes (class text embeddings) as stable anchors to guide incremental prompt tuning.

Two variants are presented:
- **TPPT-V:** Learns visual prompts guided by fixed textual prototypes, with a *textual prototypical contrastive loss* (LTPCL) that reduces representation drift.
- **TPPT-VT:** Jointly learns visual and textual prompts and introduces a *textual diversity regularization* (LDIV) to avoid feature collapse and improve uniformity.

The approach is conceptually simple, essentially prompt tuning with additional prototype-guided and diversity regularizations. Experiments on six datasets (CIFAR-100, TinyImageNet, ImageNet-R, CUB, Aircraft, Cars) show consistent improvements over previous prompt-based CL methods.

**Strengths**
- Straightforward and well-motivated design that leverages CLIP’s intrinsic textual stability.
- Strong empirical performance and extensive ablation studies (representation drift, diversity, prompt number).
- Clear writing and reproducible experimental setup.

**Weaknesses**
- Limited conceptual novelty, the method mainly combines known techniques (prompt tuning, prototype contrastive loss etc.).
- Performance appears sensitive to hyperparameters such as α, prompt depth, and number of prompts per task.
- Experiments are limited to CLIP ViT-B/16; generalization to other architectures is unclear.
- Despite good efficiency claims, the method still relies on replay buffers, which raises questions about scalability and fairness when compared to replay-free continual learning methods

**Audience:**

Yes

**Audience Explanation:**

CL for VLMs is a timely and active research area. The paper fits well within TMLR’s readership. While the conceptual novelty is moderate, the work offers a clean empirical contribution: it demonstrates how textual prototypes can stabilize multimodal CL and improve fine-grained recognition without large computational cost.

**Broader Impact Concerns:**

No major ethical issues identified. The work is methodological and low-risk. A brief note acknowledging potential bias inherited from CLIP pretraining and data reuse via replay buffers would suffice.

**Claims And Evidence:**

Yes

**Claims Explanation:**

The empirical results overall support the paper’s core claims that textual prototypes can anchor CLIP’s visual prompts to reduce representation drift, and that learning textual prompts with a diversity regularization can improve performance on fine-grained CL datasets.

However, some claims could be more rigorously supported:
- The paper frequently asserts that the method is “general” or “applicable to any model,” but all experiments are confined to CLIP ViT-B/16.
- The ablations, while extensive, still show sensitivity to hyperparameters such as the prompt depth and diversity coefficient α, which may limit reproducibility.
- The diversity regularization is motivated heuristically, with no clear theoretical backing for how it prevents collapse beyond empirical trends.
- Metrics focus on average accuracy; key continual learning measures like forward/backward transfer or forgetting are not explicitly reported.

**Requested Changes:**

- Explicitly state that the text prototype set at each incremental stage includes *only the classes seen so far*, to clarify compliance with the standard class-incremental protocol.
- Provide quantitative results for additional CL metrics (e.g., forgetting rate, backward transfer) to substantiate the claim of “reduced forgetting.”
- Provide a more intuitive explanation for how the diversity regularization improves stability beyond enforcing uniformity.
- Evaluate the method on other VLMs backbone to test generality.
- Clarify the computational overhead comparison against other methods, including training and inference cost.

---

> ### Author Response · Authors · 2025-11-06
> **Rebuttal (1/3)**
>
> We sincerely thank you for the thoughtful and constructive feedback. We appreciate that you found our method well-motivated, clearly presented, and empirically strong, and we are grateful for your detailed suggestions on broadening evaluation metrics, explaining the diversity regularization, testing additional VLM backbones, and reporting training/inference overhead. Your comments are invaluable for improving the clarity, completeness, and rigor of the paper. We address each point below and incorporate the corresponding changes in the revised manuscript.
>
> 1. **Novelty**. We appreciate the concern. TPPT builds on a standard prompt-tuning architecture (L2P, DualPrompt, CODA-P), and our core contribution is a **cross-modal, text-anchored supervision mechanism** for continual learning that (i) stabilizes the embedding space as tasks arrive and (ii) delivers consistent gains over strong prompt-based CL baselines. Importantly, we conduct continual learning on CLIP using a concise design that leverages its intrinsic anchors to mitigate forgetting, removing the need for complex and possibly unnecessary architectures or additional components used in prior work. The technical designs for this purpose are summarized below. Our results highlight the benefits of our design and its essential components.
>
> (a) **Textual prototypes.** Instead of supervising with **image class means**—a common prototypical contrastive practice that overlooks CLIP’s text modality and typically fixes prototypes after each task—TPPT uses the associated text embeddings as prototypes. This aligns naturally with CLIP’s vision–language objective, and TPPT-VT enables further refinement of these prototypes without recomputing historical class means, which would otherwise become less representative due to **representation drift** from subsequent training.
>
> (b) **Supervision with (fixed + learnable) textual prototypes.** We bring CLIP text prototypes into the CL loop as **persistent semantic anchors** that supervise visual updates across tasks, directly stabilizing the embedding space. In **Table 3**, training with these anchors via TPCL substantially improves both CIL and joint multi-task training, and this effect is also validated on the VPT baseline.
>
> \(c) **Diversity regularization on textual anchors.** We explicitly enforce uniform dispersion of text anchors on the unit hypersphere to enlarge angular margins and reduce class interference—largely overlooked by prior methods. As shown in Table 4, this regularization yields significant improvements.
>
> (d) **Concise design.** We adopt a concise incremental prompt-tuning design to validate the effectiveness of CLIP’s intrinsic textual anchors during continual learning, and it outperforms more complex prompt-tuning baselines.
>
> 2. **Robustness of hyperparameters.** Thank you for your comments. We have validated the robustness of the hyperparameters in **Figs. 7–9**. These experiments demonstrate that our method is robust to hyperparameter choices, consistently achieving satisfactory performance across a wide range of reasonable configurations.
>
> (a) **Loss weight (Fig. 7)**: we have ablated robustness across regularization weights, showing stable performance for $\alpha \in [0.5, 1.5]$; performance peaks at 0.8, though we adopt $\alpha=1.0$ by default for simplicity.
>
> (b) **Number of prompts (Fig. 8)**: using an extremely small number of prompts (i.e., 1 or 2) does show oscillatory performance, suggesting the instability of forcing broad downstream-task knowledge to be captured by very few trainable prompt tokens, which is reasonable. As long as we use a sufficient number of prompts to capture downstream-task knowledge—at least 5 prompts per task—we observe a stable performance trend, and we use 10 prompts by default to balance accuracy and cost.
>
> \(c) **Prompt depth (Figs. 9(a)–(b))**: we examine the effects of altering the depth (the number of layers to which trainable prompts are added). We observe an increasing trend when introducing prompts to deeper layers, suggesting that deeper-layer semantics are well captured by prompts added to deeper layers, bringing performance improvements.

---

> > ### Author Response · Authors · 2025-11-06
> > **Rebuttal (2/3)**
> >
> > **3. Generalization to other VLMs.** Thanks for the suggestion! To further validate effectiveness, we evaluate TPPT on **SigLIP** [A], whose pretraining differs from CLIP and yields stronger zero-shot performance. Despite these differences, TPPT, guided by stable textual anchors, **consistently improves** the SigLIP backbone across diverse downstream tasks (coarse-grained CIFAR100 and fine-grained CUB and Aircraft), achieving state-of-the-art results relative to other continual prompt tuning methods.
> >
> > [A] Zhai, X., et al. Sigmoid loss for language image pre-training. ICCV2023.
> >
> > **Table 14 in appendix of revision.** Experiment results on SigLIP.
> >
> > | Methods | CIFAR100 |  | CUB |  | Aircraft |  |
> > |:---:|:---:|:---:|:---:|:---:|:---:|:---:|
> > |  | $\bar{\mathcal{A}}$ | $\mathcal{A}_T$ | $\bar{\mathcal{A}}$ | $\mathcal{A}_T$ | $\bar{\mathcal{A}}$ | $\mathcal{A}_T$ |
> > | ZeroShot | 79.07 | 70.99 | 78.71 | 67.43 | 51.56 | 40.74 |
> > | CoOp | 77.97 | 69.44 | 75.97 | 64.29 | 48.06 | 39.63 |
> > | VPT | 85.99 | 77.90 | 80.23 | 69.80 | 51.51 | 46.28 |
> > | CODA-P | 86.69 | 79.42 | 86.11 | 80.36 | 71.33 | 69.43 |
> > | TPPT-V (Ours) | **89.16** | **81.71** | **88.18** | **82.19** | **72.00** | **69.91** |
> > | TPPT-VT (Ours) | **88.99** | **81.64** | **87.90** | **82.40** | **74.81** | **71.08** |
> >
> > 4. **Scalability and fairness of memory buffer.** Thank you for the comments! Our goal is not to design an improved memory buffer, but to study the **learning dynamics** of continual representation learning in CLIP via **textual-anchored supervision**. We adopt the standard small replay buffer to facilitate the stable learning. To assess **scalability** with many classes and long sequences, we evaluate on **DomainNet (345 classes)** split into **up to 69 tasks**. TPPT **consistently outperform** prior methods as the sequence length grows, indicating that the gains stem from textual-anchor supervision rather than memory replay. To ensure a **fair** comparison, all continual prompt-tuning baselines are run under **identical replay-buffer settings** (memory construction, same buffer size). As shown in **Table 6 in Appendix**, ablations on buffer size sweep show that TPPT’s relative gains persist across buffer sizes, highlighting the effect of our **textual-anchor objectives** (TPCL + text-side diversity) rather than **memory replay**.
> >
> >
> > **Table 11 in appendix of revision.** Comparisons on DomainNet.
> >
> > | Methods | DomainNet |  |  |  |  |  |  |  |
> > |:---:|:---:|:---:|:---:|:---:|:---:|:---:|:---:|:---:|
> > |  | 5 Tasks |  | 15 Tasks |  | 30 Tasks |  | 69 Tasks |  |
> > |  | $\bar{\mathcal{A}}$ | $\mathcal{A}_T$ | $\bar{\mathcal{A}}$ | $\mathcal{A}_T$ | $\bar{\mathcal{A}}$ | $\mathcal{A}_T$ | $\bar{\mathcal{A}}$ | $\mathcal{A}_T$ |
> > | ZS-CLIP | 63.94 | 56.66 | 63.31 | 56.66 | 63.66 | 56.66 | 63.28 | 56.66 |
> > | CoOp | 57.45 | 46.20 | 55.97 | 49.24 | 48.45 | 47.96 | 46.38 | 50.07 |
> > | VPT-CL | 68.70 | 59.42 | 70.42 | 62.78 | 68.74 | 61.24 | 71.84 | 62.49 |
> > | CODA-P | 68.04 | 59.09 | 70.68 | 61.94 | 69.72 | 61.75 | 70.45 | 61.25 |
> > | TPPT-V | **72.70** | **63.73** | **73.28** | **63.80** | **72.81** | **63.20** | **74.24** | **64.69** |
> > | TPPT-VT | **73.70** | **65.34** | **73.19** | **64.08** | **72.81** | **62.90** | **73.78** | **63.63** |
> >
> > 5. **Diversity regularization prevents collapse and improves stability.** Directly training joint vision–language embeddings risks **collapse to trivial solutions** (e.g., constant or highly redundant embeddings) without constraints; recent self-supervised objectives add explicit mechanisms to avoid this (variance/covariance [B] or redundancy penalties [C]) because naive alignment alone is insufficient. Our text-side diversity term instantiates the uniformity principle [D], by explicitly spreading textual anchors on the sphere. This balances negative forces in TPCL, **widens class margins**, and **stabilizes updates across tasks**, thereby preventing collapse and reducing correlated forgetting. Empirically, adding diversity to TPPT-VT yields consistent gains (**Table 4**, +1.42%avg / +3.31%final on CUB; +1.35%avg / +2.55%final on Aircraft).
> >
> > [B] Bardes, A., Ponce, J., & LeCun, Y. Vicreg: Variance-invariance-covariance regularization for self-supervised learning. ICLR2022.
> >
> > [C] Zbontar, J., et al. Barlow twins: Self-supervised learning via redundancy reduction. ICML 2021.
> >
> > [D] Wang, T., & Isola, P. Understanding contrastive representation learning through alignment and uniformity on the hypersphere. ICML 2020.
> >
> > 6. **Forgetting**. Thank you for the suggestion. As reported in Appendix C (Table 5), our design of learning with textual prototypes makes the method inherently less prone to forgetting. This aligns with our design: **textual prototype anchors** (fixed + learnable) provide a stable target that reduces representation drift as tasks accumulate. Notably, while **CoOp** achieves slightly lower forgetting on Aircraft, TPPT attains **substantially higher accuracy**, yielding a better overall **plasticity–stability** trade-off.

---

> > > ### Author Response · Authors · 2025-11-06
> > > **Rebuttal (3/3)**
> > >
> > > 7. **Computation cost.**
> > > As we discussed in **Sec. 4.4 and Appendix C** on  trainable-parameter counts and the compute discussion, **TPPT-V** requires **comparable or less** time because it introduces no additional learnable attention vectors, yet **delivers a +11 percentage-point gain** in final average accuracy across datasets relative to CODA-P; **TPPT-VT** additionally learns textual prompts and increases Aircraft final accuracy by **+13 percentage points**, with only **~1.5 ms** extra training time per image and **marginal increases** in peak GPU memory and FLOPs relative to TPPT-V. We here provide detailed computation costs to supplement those discussions, incluing **per-image training/inference time, peak GPU memory, and estimated FLOPs** under identical hardware, batch size, and data pipelines, comparing TPPT with related continual prompt-tuning methods.
> > >
> > >
> > > **Table 13 in appendix of revision.** Detailed computation costs. We report per-image train/inference time (ms) and peak GPU memory usage (MiB), and estimated FLOPS (GFLOPs/image).
> > >
> > > | Methods | Train |  | Inference |  | FLOPs |
> > > |---|:---:|:---:|:---:|:---:|:---:|
> > > | Methods | Time | GPU Memory | Time | GPU Memory |  |
> > > | ZS-CLIP |  |  | 2.9 | 1210.17 | 33.71 |
> > > | CoOp | 3.8 | 2171.6 | 3.2 | 1226.4 | 38.7 |
> > > | VPT | 6.2 | 5928.0 | 2.9 | 1161.6 | 37.1 |
> > > | L2P | 6.9 | 5963.3 | 2.9 | 1472.2 | 68.3 |
> > > | DualPrompt | 7.9 | 5775.1 | 2.9 | 1578.4 | 68.8 |
> > > | CODA-P | 8.2 | 7974.3 | 5.4 | 1653.7 | 68.1 |
> > > | TPPT-V (Ours) | 8.2 | 7518.1 | 5.4 | 1335.6 | 68.1 |
> > > | TPPT-VT (Ours) | 9.7 | 8821.0 | 5.9 | 1344.1 | 72.7 |
> > >
> > >
> > > **Minor**:
> > > 1. **Class coverage of prototypes.** Thanks for your suggestions! We have included the following in revision (Sec. 3.3) to explicitly clarify the coverage of classed in our text prototpyes: "At incremental step $t$, we construct the textual prototype set using only the class labels that have been introduced up to the current task. Text embeddings for future classes are not instantiated or used in any component."
> > >
> > > We thank you again for your time and effort in reviewing our paper. Your feedback was insightful and constructive. We have revised the manuscript accordingly and marked all changes and updated experimental results in blue for your reference.

---

> > > > ### Comment · Reviewer_YX6r · 2025-11-18
> > > > **Re: Rebuttal**
> > > >
> > > > Thanks to the authors for the detailed rebuttal. My concerns have been resolved, and I appreciate the improvements made to clarity and completeness in the revised manuscript.

---

> > > > > ### Author Response · Authors · 2025-11-18
> > > > >
> > > > > We are very glad to hear this. Thank you for your time and for confirming that your concerns have been resolved.

---

### Decision · Action_Editor_r7Vw · 2025-11-25

**Recommendation:** Accept as is

**Additional Comments:**

The article got initial mixed opinions, with reviewers generally appreciating the well-motivated design choices (YX6r, fq2A, vrhc), the experimental results (YX6r, fq2A, vrhc), and the presentation quality (YX6r, vrhc). However, they also raised concerns on the sensitivity to hyperparameters (YX6r), computational efficiency (YX6r, vrhc), missing discussions (fq2A), missing competitors (vrhc), and generalization to other architectures (YX6r). They also raised concerns about the technical novelty.

The response and updated manuscript addressed most of the concerns raised by the reviewers, who unanimously recommend the acceptance of the work. The AE agrees with this assessment, finding no additional changes that need to be made.

**Audience:**

Yes

**Audience Explanation:**

As the paper focuses on learning new concepts with CLIP, it is of interest to both practitioners/researchers in continual learning, and those focusing on multimodal models.

**Claims And Evidence:**

Yes

**Claims Explanation:**

The article present a method for continual learning with CLIP, exploiting textual prompts to aid learning of new classes while reducing forgetting. The effectiveness of the proposed approach is shown across multiple benchmarks (e.g., Tab. 1-2), with ablation studies (Tab. 3, Fig. 7-9) supporting the design choices.